# LEARNING CODE PREFERENCE VIA SYNTHETIC EVOLUTION

## ABSTRACT

Large Language Models (LLMs) have recently demonstrated remarkable coding capabilities. However, assessing code generation based on well-formed properties and aligning it with developer preferences remains challenging. In this paper, we explore two key questions under the new challenge of code preference learning: (i) How do we train models to predict meaningful preferences for code? and (ii) How do human and LLM preferences align with verifiable code properties and developer code tastes? To this end, we propose CODEFAVOR, a framework for training pairwise code preference models from synthetic evolution data, including code commits and code critiques. To evaluate code preferences, we introduce CODEPREFBENCH, a benchmark comprising 1364 rigorously curated code preference tasks to cover three verifiable properties—correctness, efficiency, and security—along with human preference. Our evaluation shows that CODEFAVOR holistically improves the accuracy of model-based code preferences by up to $28.8\%$. Meanwhile, CODEFAVOR models can match the performance of models with $6 \sim 9\times$ more parameters while being $34\times$ more cost-effective. We also rigorously validate the design choices in CODEFAVOR via a comprehensive set of controlled experiments. Furthermore, we discover the prohibitive costs and limitations of human-based code preference: despite spending 23.4 person-minutes on each task, $15.1 \sim 40.3\%$ of tasks remain unsolved. Compared to model-based preference, human preference tends to be more accurate under the objective of code correctness, while being sub-optimal for non-functional objectives.

## 1 INTRODUCTION

Large Language Models (LLMs) for code (Chen et al., 2021; GitHub, 2023; Amazon Web Services, 2023) have become instrumental in modern software development. Code LLMs assist developers in various scenarios, from suggesting code completions and generating functional code based on user instructions to proposing complex code changes to resolve bug reports and feature requests.

Instruction-tuned LLMs (Luo et al., 2024; Wei et al., 2024) are increasingly adept at generating functional code based on natural language instructions. However, evaluating the quality of LLM-generated code remains challenging, particularly regarding code correctness, efficiency, security, adherence to best practices, and alignment with developer preferences. Effectively and efficiently assessing LLM-generated code against these properties is crucial for both evaluation (Liu et al., 2023b) and preference optimization for code LLMs (Weyssow et al., 2024). Nevertheless, the subject of learning code preferences has been largely under-explored, motivating us to study code preferences systematically and train code preference models with new data and modeling methods.

Following the established format in LLM-as-a-judge (Chiang et al., 2024), we define the code preference task as follows: Given a user query, a pair of two candidate code responses, and optionally a preference criterion, code preference is demonstrated by choosing one response over the other. Specifically, current approaches estimate code preference based on three proxies, each with advantages and limitations:

- **Code execution:** Code preference in another way can be confidently determined by execution statuses (Liu et al., 2023a). However, applying code execution to arbitrary programs poses challenges due to *(i)* setup complexity, *(ii)* code incompleteness, and *(iii)* execution overhead. For instance, code execution may necessitate specific hardware (*e.g.,* GPUs) and precise software versions, which are challenging to deduce from the code and, even if inferred, are too cumbersome to set up and run.
- **Human annotation:** Human-labeled preferences are often seen as the standard oracle in developing LLMs, such as in the RLHF for OpenAI's GPT models (Ouyang et al., 2022) and LLM evaluation

in Chatbot Arena (Chiang et al., 2024). However, applying human labeling to code is particularly challenging and cost-intensive. Programs are inherently abstract and complex, labeling them requires experienced developers to perform detailed analysis and testing. Meanwhile, human preference is inherently subjective, influenced by the annotators' code tastes and expertise, which can cause noisy preferences for code, whose quality could otherwise be concretely defined and measured.

- **LLM-as-a-judge:** Prominent LLMs have also been employed to evaluate LLM responses (Chiang et al., 2024; Zheng et al., 2023; McAleese et al., 2024). This method is more scalable than human labeling and can be generalized to a wider range of programs compared to code execution. However, its reliability often hinges on the reasoning capabilities of high-cost proprietary LLM judges (Weyssow et al., 2024), subject to inherent biases (Zheng et al., 2023).

While scaling human- and execution-based preference for code is human-resource- and engineering-challenging[1], improving model-based code preference becomes emerging and crucial, beyond directly prompting off-the-shelf models (Weyssow et al., 2024). Furthermore, how exactly human developers and prominent LLMs determine code preference remains obscure, with little research on quantifying or analyzing their performance across various code criteria. To this end, this work attempts to explore two critical questions in code preference learning:

1. **Technical question:** How can we build effective and efficient code preference models regarding modeling approaches and data sources?
2. **Empirical question:** What are the preferences of human annotators and LLMs, and to what extent do they align with verifiable code properties and human judgments?

**CODEFAVOR.** We propose CODEFAVOR, a novel framework for training code preference models. Specifically, CODEFAVOR employs pairwise modeling to predict preference within a code pair according to a user-specified criterion. We propose two synthetic data generation methods to construct preference ranking samples from code evolution: *(i) Commit-Instruct* transforms the pre- and post-commit code snippets to code preference pairs; and *(ii) Critic-Evol* samples faulty code from a draft LLM and has another critic LLM to improve the broken code. These methods allow us to curate synthetic preference data efficiently, leveraging the natural evolution of code and the capabilities of existing LLMs.

**CODEPREFBENCH.** To evaluate code preferences labeled by various approaches, we introduce CODEPREFBENCH, a collection of 1,364 carefully curated preference tasks. These tasks target verifiable properties including correctness, efficiency, and security, while additionally considering general developer preferences. Using CODEPREFBENCH, we extensively analyze the effectiveness and cost of code preferences derived from developer agreement, general LLMs, and CODEFAVOR models. Our study demystifies key insights on the pitfalls of different approaches over different coding criteria. Our results also demonstrate that our models not only achieve top performance in effectiveness but also are significantly more cost-efficient compared to existing solutions.

We summarize our main contributions below:

1. **Dimension & Technique:** We propose CODEFAVOR, the *first* open recipe to train pairwise code preference models. At the heart of CODEFAVOR is a pairwise modeling design and two complementary methods for generating synthetic preference pairs from code evolution.
2. **Benchmark & Code:** We present CODEPREFBENCH, the *first* comprehensive developer preference benchmark with 1,364 labeled by three verifiable oracles (correctness, efficiency, security) and general developer preferences from 18 annotators. We release the data and code at `[ANONYMOUS-URL-FOR-PEER-REVIEW]`.
3. **Study & Results:** Based on CODEPREFBENCH, we comprehensively quantify and conduct case studies on code preferences derived from human developers and LLMs. We show that CODEFAVOR can significantly improve the accuracy of model-based preference by up to 28.8%. CODEFAVOR models can match the preference accuracy of models that are larger by $6 \sim 9\times$, while being cheaper by $34\times$. We also conduct extensive controlled experiments to validate our design choices.

## 2 CODEFAVOR: LEARNING CODE PREFERENCE VIA SYNTHETIC EVOLUTION

Figure 1 depicts the approach overview. Our framework, CODEFAVOR, is designed for training pairwise models that predict code preference by taking an instruction, a code pair, and a criterion as input.

---

[1]Such as hiring more annotators with domain expertise and setting up individual execution environments.

Figure 1: Approach overview of CODEFAVOR. We train a pairwise preference model using synthetic data created from two complementary sources of code evolution: *Commit-Instruct* and *Critic-Evol*.

Additionally, CODEFAVOR proposes two synthetic data generation methods, *Commit-Instruct* and *Critic-Evol*, for extracting synthetic training data from code evolution. Specifically, *Commit-Instruct* creates contrasting code pairs through rephrasing and filtering massively available code commits. Complementarily, *Critic-Evol* prompts a large critic LLM to judge and revise code snippets from a smaller draft LLM, pairing the drafted attempt and revision to create synthetic preference data.

## 2.1 PAIRWISE MODELING

**Input.** We follow prior work in reward modeling (Zhao et al., 2023; Liu et al., 2024b; Dong et al., 2024) and use decoder-based transformers for learning pairwise preferences. Specifically, the model $\pi$ takes as input a prompt $x = \{i, y_A, y_B, c\}$, comprised of *(i)* an instruction $i$, *(ii)* a pair of code candidates $\{y_A, y_B\}$, and *(iii)* a fine-grained criterion $c$ defining the desired preference following (Kim et al., 2023). More specifically, our prompt format is shown in Equation (1):

$$x = [\texttt{INSTRUCTION}]\{i\}[\texttt{CODE\_A}]\{y_A\}[\texttt{CODE\_B}]\{y_B\}[\texttt{CRITERION}]\{c\} \tag{1}$$

**Output.** We explore two output designs for code preference modeling: classification and generation.

1. *Classification:* We train a binary classifier based on a single next-token prediction (Zhao et al., 2023; Liu et al., 2024b). Specifically, given the exact prompt format in Equation (1) formatted by a chat template, the classifier outputs either a token "A" if $y_A$ is preferable to $y_B$ for $\{i, c\}$ or "B" otherwise. Notably, the single output token is separated from the prompt by special tokens defined by the chat template. At inference time, the preference decision is determined by the next-token probability between "A" and "B", shown in Equation (2).
2. *Generation:* We also train generative models to provide code preference decisions in natural language. Specifically, we use a more natural version of Equation (1), demonstrated in Listing 1, for prompting the generation model. Next, we parse the code preference decision from the model-generated feedback using rules detailed in Appendix A.3.

$$y_+ = \begin{cases} y_A & \text{if } \mathbb{P}_\pi(y_A \succ y_B | x) > 0.5 \\ y_B & \text{otherwise} \end{cases} = \begin{cases} y_A & \text{if } \mathbb{P}_\pi(\text{``A''}|x) > \mathbb{P}_\pi(\text{``B''}|x) \\ y_B & \text{otherwise} \end{cases} \tag{2}$$

The advantage of classification modeling is computing efficiency as only one token is produced. Meanwhile, generative modeling optimizes for interpretability, with reasoning steps explicitly displayed.

## 2.2 SYNTHETIC CODE PREFERENCE FROM CODE EVOLUTION

Training a pairwise preference model requires a rich set of contrastive code pairs, along with the corresponding instructions and evaluation criteria. Collecting complex (Luo et al., 2024) and diverse (Wei et al., 2024) code pairs is crucial yet challenging, given such resources are neither readily available nor curated by prior work. To this end, we propose to create code preference training data using synthetic code evolution, based on code commits (§2.2.1) and code critiques (§2.2.2). We argue that code evolution is a practical source for synthesizing code preferences, not only because of its natural

indication of preferences[2], but also thanks to their general availability and diversity. We focus on the general methodology in this section and defer the detailed prompting implementation to Appendix A.1.

### 2.2.1  *Commit-Instruct*: PREFERENCE LEARNING FROM CODE COMMITS

We propose *Commit-Instruct*, a synthetic data generation method transforming raw code commits into code preference training samples. Specifically, the workflow (middle of Figure 1) employs a critic LLM to analyze each raw code commit and produce a training sample in a desired format §2.1. Each raw commit can be denoted by $r = (m, y_{pre}, y_{post})$, where $m$ is the commit message, and $\{y_{pre}, y_{post}\}$ are the pre- and post-commit code snippets. *Commit-Instruct* processes each commit in three steps:

1. **Reasoning:** The critic LLM is instructed to reason and explain code changes from $y_{pre}$ to $y_{post}$.
2. **Filtering:** Given the explanation, the critic LLM first determines whether or not the code change is meaningful. If so, we proceed to the next step; otherwise, the commit is discarded. This step aims to ensure the quality of synthetic data by excluding trivial or ambiguous code changes.
3. **Rephrasing:** Based on the commit $r$ and its explanation, the critic LLM synthesizes a preference sample in the desired format $x = \{i, y_A, y_B, c\}$ (§2.1). Specifically, $y_A$ and $y_B$ are rephrased from $y_{pre}$ and $y_{post}$ to emphasize the actual change. $i$ is the instruction generated to describe $y_{pre}$ and $y_{post}$ and the criterion $c$ is concluded by how $y_{post}$ improves $y_{pre}$. The rephrased version of $y_{post}$ is regarded as the chosen response $y^+$ in model training.

Figure 3 in Appendix A.1 provides the detailed prompt implementation for *Commit-Instruct*.

### 2.2.2  *Critic-Evol*: PREFERENCE LEARNING FROM CODE CRITIQUES

While synthetic evolution can be gathered from human data such as code commits, it can also be derived from synthetic data. As such, we propose *Critic-Evol* which generates synthetic code preference data by asking a stronger *critic* model $\pi^+$ to revise the code generated by a weaker *draft* model $\pi^-$.

Specifically, *Critic-Evol* takes a set of coding instructions $\mathcal{I} = \{i_1, i_2, \cdots, i_n\}$ as inputs, for each of which $(i_k)$ we apply steps below to either transform it into a synthetic training sample or simply drop it:

1. **Fault sampling:** This step starts with a suitably weak but functional model, denoted as $\pi^-$, which statistically allows us to sample improvable code solutions $y_k^- \leftarrow \pi^-(i_k)$.
2. **Critique & filtering:** We instruct the critic LLM $\pi^+$ to code review $y_k^-$, by pointing out noticeable code quality issues and defining the criterion (*e.g.,* $c$) regarding the code defects. $\pi^+$ may also be satisfied with $y_k^-$ and thus we stop synthesizing code preference data for $(i_k, y_k^-)$.
3. **Revision:** If the critique from $\pi^+$ suggests $y_k^-$ can be significantly improved, $\pi^+$ creates $y_k^+$ by revising $y_k^-$ to meet the desired criterion $c$. As such, a new synthetic code preference sample is composed as $\{i_k, y_k^-, y_k^+, c\}$, with $y_k^+$ being the chosen response.

Figure 5 in Appendix A.1 provides more details on implementation of *Critic-Evol*.

### 2.3  DATASETS

We apply our techniques to create two synthetic datasets for code preference learning:

***Commit-Instruct*-EditPack** consists of 20,641 code preference samples synthesized from EditPackFT-Multi (Cassano et al., 2023) and Llama3-70B-Instruct (Dubey et al., 2024). After filtering out non-permissive code, we obtain 22,469 blessed Python commits from EditPackFT-Multi for use as the raw commits and prompt Llama3-70B-Instruct (Dubey et al., 2024) to perform the *Commit-Instruct* strategy. 91.9% of the commits are successfully transformed into code preference data (§2.1) and 8.1% of them are filtered out due to lack of clear significance.

***Critic-Evol*-SOSS** has 41,595 synthetic code preference samples using the *Critic-Evol* technique. Specifically, we run Llama3-8B-Instruct as the draft model (*i.e.,* $\pi^-$) over 50,661 coding instructions from the Self-OSS-Instruct dataset (BigCode, 2024) to produce initial code solutions. 82.1% of these initial attempts are revised and extended by Llama3-70B-Instruct as the critic model, whereas the rest 17.9% are deemed good enough such that a revision is unnecessary.

---

[2]Software iterates towards improvement, making post-evolution code oftentimes better than pre-evolution code.

| Objective | # Tasks | Source | Preference Oracle |
|---|---|---|---|
| Code Correctness | 660 | EvalPlus (Liu et al., 2023b) | Test execution |
| Code Efficiency | 352 | EvalPerf (Liu et al., 2024a) | # CPU instructions |
| Code Security | 207 | CyberSecEval (Bhatt et al., 2023) | Static analyzer |
| Developer Preference | 145 | LBPP (Matton et al., 2024) BigCodeBench-Hard (Zhuo et al., 2024) | Human agreement |
| **Total** | 1,364 | | |

Table 1: Overview of CODEPREFBENCH.

**Data processing.** To mitigate positional bias, we augment the dataset by flipping the order within each code pair, which also doubles the training samples. Besides, we clip the code comments in *Critic-Evol* samples, given that comments barely affect code quality metrics and LLM-generated comments may let faulty code "sound right". §3.4 also shows code comments can negatively impact code preferences.

## 3 EVALUATING CODE PREFERENCE LEARNING WITH CODEPREFBENCH

To systematically evaluate code preferences across different methods, we create the CODEPREFBENCH, a set of 1,364 preference tasks. It covers four objectives in code preference evaluation: correctness, efficiency, security, and human preference. Table 1 provides an overview of the four categories of tasks.

This section presents the curation process of CODEPREFBENCH (§3.1) and the results from human (§3.2) and LLMs (§3.3), along with the controlled experiments in §3.4. Additional details, such as case studies (Appendix A.4) and contamination analysis (Appendix A.5), are deferred to the Appendix.

### 3.1 BENCHMARK SETUP

In CODEPREFBENCH, we evaluate code preference approaches over four objectives, covering three verifiable properties (*i.e.,* correctness, efficiency, and security) and human preference. For verifiable objectives, we generate oracle labels via code execution and static analysis. For human preference, we engage three annotators to label each code pair to form the evaluation set and establish baselines To ensure benchmark quality, we only use clear-cut *good-bad* pairs and exclude *tie* pairs due to their inherent ambiguity. The creation of the dataset for each evaluation category is detailed below:

**Objective #1: Correctness.** We construct *correct-wrong* pairs from EvalPlus datasets (Liu et al., 2023b), *i.e.,* HumanEval+ (164 tasks) and MBPP+ (378 tasks), as they rigorously test LLM solutions with extensive test cases that can detect subtle bugs. We derive at most two contrastive code pairs for evaluation from each seed task. In each code pair, the *wrong* code comes from test-falsified LLM solutions while the *correct* is the human-written ground truth. Finally, we obtain 660 *correct-wrong* code pairs. The number is smaller than $2 \times (164 + 378)$ as "wrong" samples do not exist in some easy tasks.

**Objective #2: Efficiency.** We construct *fast-slow* pairs from EvalPerf datasets (Liu et al., 2024a). EvalPerf exercises the performance of LLM-generated correct solutions using 121 performance-exercising tasks equipped with performance-exercising test inputs. The EvalPerf dataset provides fast-to-slow reference solutions with distinct performance for each task. Therefore, we sample *fast-slow* pairs over the reference samples at a step size of 3, and obtain 352 *fast-slow* pairs.

**Objective #3: Security.** We construct *secure-vulnerable* code pairs from CyberSecEval (Bhatt et al., 2023), which includes 351 Python vulnerabilities detected by security analyzers. We prompt GPT-4o to fix each vulnerability and rerun the security analyzers to guarantee the fix. Additionally, we equip each code pair with a *generalized* instruction generated by GPT-4o, so the instruction is not biased towards any candidate. Finally, we obtain 207 *secure-vulnerable* code pairs to evaluate code security preference.

**Objective #4: Human preference.** We established a team of 18 developers to annotate pairs of code responses sampled from DeepSeek V2 over the latest open and close domain coding benchmarks, *i.e.,* 148 BigCodeBench-Hard (Zhuo et al., 2024) tasks and 161 LBPP (Matton et al., 2024) tasks. Specifically, we sample 8 solutions per task at a temperature of 0.8 and select the code pair with the largest edit distance. We follow the same annotation criteria as Chatbot Arena (Chiang et al., 2024): given

two responses, users select the one they would use for the instruction (or skip it if both are tied). Lastly, we obtained 145 preference pairs without conflicting preferences out of three annotations per pair.

Additionally, we evenly shuffle the order of code pairs within each category to prevent positional bias. By default, we remove code comments when evaluating tasks focused on verifiable objectives, as comments should not affect the outcome. At evaluation, LLMs predict each code preference task using greedy decoding, following criteria aligned with the benchmark objective.

## 3.2 HUMAN RESULTS

We first study and draw insights from developer labeling through their expertise, annotation confidence, annotation time, and finally their results based on the major voting of 3 developers per task:

- **Expertise:** Our annotation team consists of 18 software developers, two-thirds of which hold degrees in computer science, and 95% of them have over two years of programming experience. For Python proficiency, 43% of them self-rate as advanced, while the rest consider themselves middle-level.

|  | Low | High | Very High |
|---|---|---|---|
| Correctness | 0% | 68.2% | 31.8% |
| Efficiency | 0% | 88.7% | 11.3% |
| Security | 0% | 80.8% | 19.2% |

Table 2: Developer confidence distribution.

- **Confidence:** Table 2 lists the distribution of developer confidence. All developers are overall confident about their annotations. Specifically, developers are more confident when labeling correctness, with a higher ratio of *"very high"* confidence compared to that for the efficiency ($2.8\times$) and security ($1.7\times$) categories. From annotation notes of developers, it is partially because program correctness can be assessed by manual testing, while code efficiency and security are harder to evaluate without domain-specific knowledge.

- **Overhead:** Figure 2 illustrates the cumulative distribution of the annotation time per sample/developer, visualized by removing the top-1%-longest outliers. Overall, each task on average costs each developer 7.8 minutes to annotate, with the 99-percentile of 26 minutes, indicating that serious developer labeling for code preferences is time-consuming. Furthermore, code efficiency and security tasks (9 minutes on avg.) take longer to annotate compared to labeling correctness tasks (6.8 minutes on avg.), which is consistent with developer confidence and final results.

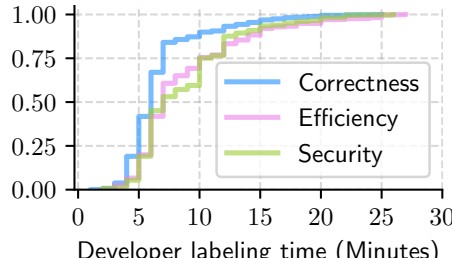

Figure 2: CDF of preference labeling time.

- **Accuracy:** The first result row in Table 3 presents the human performance based on the majority voting of three developers per task. Consistent with the annotation confidence and speed, human labeling achieves the most accurate measurement of code preferences in the code correctness category, with a solve rate of 84.9%. While not the best, human performance still decently solves 74.9% of preference tasks targeting code efficiency. Surprisingly, while developer confidence in code security annotation is higher than that in code efficiency, the security score is as low as 59.7%. This is because 73.9% of code pairs are annotated as equally secure, while our scoring method assigns 0.5 accuracy to each tied case. This discrepancy indicates that generalist programmers may struggle to accurately assess non-functional code properties such as code security, which may require specialized domain expertise.

## 3.3 MODEL RESULTS

Table 3 evaluates human, existing LLMs, and CODEFAVOR models on CODEPREFBENCH. By default, CODEFAVOR models are obtained in two steps: *(i)* training two models using *Commit-Instruct*-EditPack and *Critic-Evol*-SOSS individually; and *(ii)* merging the two models on average to obtain a final model.

**Overall results.** We present the overall results by looking at the accuracy averaged across the three verifiable objectives, *i.e.,* the"**Avg.**" column. Among the evaluated existing LLMs, Llama-3.1-405B-Instruct and Mistral Large 2 perform the best, tightly followed by Claude 3.5 Sonnet and DeepSeek V2.5. Meanwhile, Codestral, at a parameter size of 22B, demonstrates a decent result, on par with Llama-3-70B-Instruct. We demonstrate the effectiveness of CODEFAVOR by fine-tuning a comprehensive set of affordable models, from 7B to 12B. While these small models are relatively weak out of the box, CODE-FAVOR improves their overall performance by $9.3\sim28.8\%$ relatively. For instance, CODEFAVOR's generation modeling enables Mistral Nemo Instruct, Gemma-2-9B-Instruct, and Llama-3-8B-Instruct to

| | Correctness | Efficiency | Security | Avg. | Human Pref. |
|---|---|---|---|---|---|
| 3-developer agreement | 84.9 ($\pm 9.4$) | 74.9 ($\pm 5.3$) | 59.7 ($\pm 37.0$) | 73.2 | N/A |
| Logprob Mean | 32.4 | 38.4 | 59.9 | 43.6 | 42.1 |
| **Proprietary Models** | | | | | |
| Claude 3.5 Sonnet | 65.8 ($\pm 0.8$) | 79.9 ($\pm 0.1$) | 98.1 | **81.2** | 64.8 |
| Gemini 1.5 Pro 001 | 59.2 ($\pm 3.0$) | 79.5 ($\pm 1.4$) | 71.3 ($\pm 27.3$) | 70.0 | 66.6 ($\pm 1.7$) |
| Gemini 1.5 Flash 001 | 58.6 ($\pm 7.9$) | 81.1 ($\pm 0.1$) | 85.0 ($\pm 8.2$) | 74.9 | 60.0 |
| **Open-Weight Models** | | | | | |
| Llama-3.1-405B-Instruct | 68.9 ($\pm 2.7$) | 78.3 ($\pm 0.4$) | 99.0 | **82.2** | 68.3 |
| Mistral Large 2 (123B) | 65.8 ($\pm 0.5$) | 81.2 ($\pm 0.3$) | 99.5 | **82.2** | 71.7 |
| DeepSeek V2.5 (236B) | 65.8 ($\pm 0.8$) | 80.7 | 97.3 ($\pm 0.2$) | **81.3** | 69.0 |
| Llama-3.1-70B-Instruct | 60.2 ($\pm 0.3$) | 77.3 ($\pm 0.3$) | 97.8 ($\pm 0.7$) | 78.4 | 69.0 |
| Codestral-22B-v0.1 | 58.0 ($\pm 0.8$) | 78.3 ($\pm 0.1$) | 94.0 ($\pm 2.7$) | 76.8 | 60.0 |
| Llama-3-70B-Instruct | 55.7 ($\pm 2.5$) | 76.0 ($\pm 1.6$) | 96.6 ($\pm 1.0$) | 76.1 | 63.8 ($\pm 0.3$) |
| Gemma-2-27B-Instruct | 55.4 ($\pm 4.9$) | 78.4 ($\pm 0.9$) | 80.8 ($\pm 14.8$) | 71.5 | 61.4 |
| Skywork-Reward-Llama-3.1-8B | 56.2 | 64.2 | 61.4 | 60.6 | 57.9 |
| **Our Models and Baselines** | | | | | |
| Mistral Nemo Instruct (12B) | 51.4 ($\pm 1.2$) | 69.7 ($\pm 0.4$) | 82.9 ($\pm 7.5$) | 68.0 | 66.2 |
| + CODEFAVOR Classification | 58.0 | 76.1 | 96.6 | **76.9** | 64.1 |
| + CODEFAVOR Generation | 58.8 | 77.8 | 96.6 | **77.7** | 66.9 |
| Gemma-2-9B-Instruct | 52.4 ($\pm 6.1$) | 75.1 ($\pm 1.6$) | 52.7 ($\pm 47.3$) | 60.1 | 64.1 ($\pm 0.7$) |
| + CODEFAVOR Classification | 56.8 | 75.3 | 92.3 | 74.8 | 67.6 |
| + CODEFAVOR Generation | 57.0 | 78.7 | 96.6 | **77.4** | 64.1 |
| Llama-3-8B-Instruct | 49.5 ($\pm 0.9$) | 71.9 | 90.3 ($\pm 0.5$) | 70.6 | 58.6 |
| + CODEFAVOR Classification | 58.0 | 73.0 | 95.2 | 75.4 | 62.8 |
| + CODEFAVOR Generation | 58.2 | 75.0 | 98.6 | **77.2** | 69.0 |
| Mistral-7B-Instruct-v0.3 | 48.5 ($\pm 1.5$) | 66.6 ($\pm 0.1$) | 78.5 ($\pm 9.4$) | 64.5 | 58.3 ($\pm 1.0$) |
| + CODEFAVOR Classification | 62.4 | 64.8 | 95.7 | **74.3** | 60.7 |
| + CODEFAVOR Generation | 57.1 | 77.3 | 90.3 | **74.9** | 66.9 |

Table 3: Preference accuracy (%) on CODEPREFBENCH. The best score and scores within 1 percentage point of the best score are highlighted in **bold**. Bracketed numbers denote the ranges of uncertain responses ("Tie"), half of which account for the final score. Appendix A.4 presents some case studies.

achieve an overall score of $77.2 \sim 77.7$ respectively, slightly outperforming the critic model (*i.e.,* Llama-3-70B-Instruct), despite being smaller by $6 \sim 9 \times$. Meanwhile, we show that CODEFAVOR models are even better than the best 8B general reward model on RewardBench (Lambert et al., 2024), *i.e.,* Skywork-Reward-Llama-3.1-8B-v0.2, by 24% on CODEPREFBENCH, even if the compared reward model is trained on a rich set of data generated by proprietary models such as GPT-4 and Claude-3-Opus. Notably, all CODEFAVOR models even outperform the human-agreement baseline, largely because generalist developers have high uncertainty and thus low performance in the security category. Besides model- and human-based approaches, we also evaluate selecting preferred samples using the mean log probabilities of decoding Llama-3.1-8B-Instruct, which overall performs randomly on CODEPREFBENCH.

**Correctness.** Human annotation largely outperforms all language models in choosing the correct code, outperforming the best model by 23%. Among the evaluated existing LLMs, Llama-3.1-405B-Instruct as an open-weight model solves the most tasks (*i.e.,* 68.9%), outperforming Claude 3.5 Sonnet, Mistral Large 2, and DeepSeek V2.5 by 4.7%. Meanwhile, small LLMs ($\leq 12$B) are incapable of such tasks out of the box, producing almost random preferences ($\sim 50\%$). Nonetheless, CODEFAVOR improves the accuracy of code correctness preference for these models by $8.8 \sim 28.7\%$, commonly surpassing their critic model (*i.e.,* Llama-3-70B-Instruct) by up to 12%.

**Efficiency.** While human preference aces over the evaluated LLMs on the preference of code preference, it presents overall sub-optimal preferences regarding code efficiency. For example, Mistral Large 2, the best model in this category, surpasses developer-agreement-based preference by $8.4\%$. Gemini Flash and DeepSeek V2.5 tightly follow Mistral Large 2 within a 1% gap, also outperforming human preference in the code efficiency category. While smaller LLMs perform more decently compared to their results on code correctness preference, CODEFAVOR still further improves them by up to $16.1\%$, on par with or slightly surpassing the critic LLMs (by up to 4%).

**Security.** The code security subset in CODEPREFBENCH is relatively much easier given that most models achieve saturated scores, *e.g.,* Mistral Large 2 solves 99.5% tasks. Similar to the low performance of the human baseline, the scores of Gemini 1.5 Pro and Gemma 2 models are surprisingly low, with up to 47.3% of code pairs regarded as equally insecure (*e.g.,* Figure 13), even if the evaluation prompt in Listing 1 asks for an absolute answer. Nevertheless, such behavior can be benign for mitigating potential security risks from deceptive prompts with the actual purpose of assisting cyber-security attacks. Meanwhile, small models are still improvable in this dimension. For example, CODEFAVOR eliminates the uncertainty in Gemma-2-9B-Instruct and improves its security preference score by up to 89%. For other small models, CODEFAVOR can still improve them by $9.2 \sim 21.9\%$.

**Human preference.** Aligning the objective of human preference is as challenging as that in the correctness category. The best model, Mistral Large 2, can solve 70-ish percent of tasks, outperforming the best-evaluated proprietary model, Gemini 1.5 Pro, by 7.7%. While in human preference smaller LLMs perform much better than in the correctness objective, CODEFAVOR can still improve their performance in aligning human preference for code by up to 17.7%. By checking the dataset, the difficulty of aligning human preferences can partially come from the ambiguity and bias inherent in human annotation. Sometimes both code candidates exhibit different advantages and disadvantages, making human preference just one of many possible reasonable judgments, rather than the definitive answer.

**Preference cost.** Besides preference accuracy, Table 4 lists the cost of representative approaches for running CODEPREFBENCH. Specifically, human agreement as the most expensive approach costs $6.1 per task, estimated by their annotation time and California's minimal wage. Llama-3.1-405B-Instruct, with the best overall performance, is two orders of magnitude cheaper than human preference. While Llama-3-70B-Instruct is 7.4% weaker than the 405B model, it is cost-effective for being cheaper by $35.3\times$. Furthermore, CODEFAVOR models present the best cost-effectiveness. CODEFAVOR classifier based on Mistral Nemo Instruct is five orders of magnitude cheaper than human preference and is $34\times$ cheaper than Llama-3-70B-Instruct, with no worse preference accuracy.

|  | Norm. Cost | Accuracy |
|---|---|---|
| Human preference ($3\times$) | $1.2\times10^5$ ($6.1) | 73.2 |
| Llama-3.1-405B-Instr. | $1.2\times10^3$ | 82.2 |
| Llama-3-70B-Instr. | $3.4\times10^1$ | 76.1 |
| Ours (Mistral Nemo) | 1 | 76.9 |

Table 4: Estimated per-sample cost and accuracy.

## 3.4 CONTROLLED EXPERIMENTS

This section (and Appendix A.6) studies the design choices in CODEFAVOR via controlled experiments.

**Training data.** As we have two sources of training datasets, in Table 5 we study their training effect individually (*i.e., "Commit-Instruct"* and *"Critic-Evol"*) and in combination (*i.e., "Data Mixture"*). Comparing *Commit-Instruct* with *Critic-Evol*, models trained by *Critic-Evol* tend to achieve better overall performance, particularly in the correctness category. For example, within the classifier modeling, *Critic-Evol* surpasses *Commit-Instruct* by $6.1 \sim 15.5\%$ on the correctness category, and the overall improvement (*i.e., "Avg."* column) can be up to 7.5%. Meanwhile, when using the classification modeling, *Commit-Instruct*-enabled models can perform better in the preference for code efficiency, with up to 12.5% improvement. Moreover, data mixture can further improve the effectiveness of model-based preference, especially when using generation modeling, with up to 8.7% and 4.3% improvement over *Commit-Instruct* and *Critic-Evol* respectively. The performance trend correlates with the training sample sizes, indicating that the more training data, the better performance.

**Data mixture *v.s.* model merging.** In addition to data mixture, we also explore co-utilizing both training datasets via model merging (Wortsman et al., 2022), by averaging the weights of two models trained by individual datasets. Model merging yields better evaluation results for all trained classification models, with $1.1 \sim 5.0\%$ improvements. Within the generation modeling, model merging also surpasses or stays on par with data mixture results for all model types except for the Mistral 7B series.

**Classification *v.s.* generation.** Table 5 also compares the output representation between classification and generation. One qualitative trend is that classifier modeling often leads to higher scores in the preference for code correctness while the generation modeling tends to bring more holistic improvement leading to a higher overall store. For example, within the 16 comparisons in Table 5, the classification modeling outperforms the generation modeling 9 times in the code correctness objective, whereas the generation modeling surpasses the classification modeling 13 times in the average score.

| | | Correctness | Efficiency | Security | Avg. | Human Pref. |
|---|---|---|---|---|---|---|
| **Mistral Nemo Instruct (12B)** | | 51.4 ($\pm$1.2) | 69.7 ($\pm$0.4) | 82.9 ($\pm$7.5) | 68.0 | **66.2** |
| *Commit-Instruct* | Classification | 54.5 | **79.3** | **96.1** | 76.6 | 65.5 |
| | Generation | 48.2 | 74.4 | **96.6** | 73.1 | **66.9** |
| *Critic-Evol* | Classification | 59.8 | 70.5 | **95.7** | 75.3 | 62.1 |
| | Generation | 56.8 | 77.0 | **96.6** | **76.8** | 61.4 |
| Data Mixture | Classification | **63.0** | 68.8 | 95.2 | 75.6 | 62.1 |
| | Generation | 58.2 | 77.0 | **96.1** | **77.1** | 64.1 |
| Model Merging | Classification | 58.0 | 76.1 | **96.6** | **76.9** | 64.1 |
| | Generation | 58.8 | 77.8 | **96.6** | **77.7** | 66.9 |
| **Gemma-2-9B-Instruct** | | 52.4 ($\pm$6.1) | 75.1 ($\pm$1.6) | 52.7 ($\pm$47.3) | 60.1 | 64.1 ($\pm$0.7) |
| *Commit-Instruct* | Classification | 52.3 | 71.9 | 82.1 | 68.8 | 63.4 |
| | Generation | 51.8 | **80.1** | 95.1 | 75.3 | 60.7 |
| *Critic-Evol* | Classification | 55.5 | 74.7 | 86.5 | 72.2 | 62.1 |
| | Generation | 57.9 | 72.2 | **97.6** | 75.9 | 64.1 |
| Data Mixture | Classification | 54.8 | 73.9 | 87.9 | 72.2 | 63.4 |
| | Generation | **59.2** | 76.7 | **97.6** | **77.8** | 63.4 |
| Model Merging | Classification | 56.8 | 75.3 | 92.3 | 74.8 | **67.6** |
| | Generation | 57.0 | 78.7 | 96.6 | **77.4** | 64.1 |
| **Llama 3-8B-Instruct** | | 49.5 ($\pm$0.9) | 71.9 | 90.3 ($\pm$0.5) | 70.6 | 58.6 |
| *Commit-Instruct* | Classification | 54.4 | 71.0 | 93.7 | 73.0 | 65.5 |
| | Generation | 48.9 | 73.0 | 94.2 | 72.1 | 66.2 |
| *Critic-Evol* | Classification | **58.3** | 71.3 | 90.3 | 73.3 | 57.9 |
| | Generation | **58.3** | **74.4** | 93.7 | 75.5 | **69.0** |
| Data Mixture | Classification | **58.5** | 66.2 | 90.8 | 71.8 | 62.1 |
| | Generation | 56.8 | 73.6 | 94.7 | 75.0 | 66.2 |
| Model Merging | Classification | **58.0** | 73.0 | 95.2 | 75.4 | 62.8 |
| | Generation | **58.2** | 75.0 | **98.6** | **77.2** | **69.0** |
| **Mistral-7B-Instruct-v0.3** | | 48.5 ($\pm$1.5) | 66.6 ($\pm$0.1) | 78.5 ($\pm$9.4) | 64.5 | 58.3 ($\pm$1.0) |
| *Commit-Instruct* | Classification | 55.5 | 69.3 | 83.1 | 69.3 | 61.4 |
| | Generation | 48.0 | 73.3 | 88.4 | 69.9 | **66.2** |
| *Critic-Evol* | Classification | **64.1** | 64.8 | 94.7 | 74.5 | 61.4 |
| | Generation | 57.7 | 72.4 | 88.4 | 72.9 | 58.6 |
| Data Mixture | Classification | 59.5 | 69.3 | 91.8 | 73.5 | 60.7 |
| | Generation | 61.7 | 73.6 | 92.8 | **76.0** | 62.8 |
| Model Merging | Classification | 62.4 | 64.8 | **95.7** | 74.3 | 60.7 |
| | Generation | 57.1 | **77.3** | 90.3 | 74.9 | **66.9** |

Table 5: CODEPREFBENCH results of CODEFAVOR models using different training data and modeling.

**Criteria.** Table 6 studies the impact of criteria in the prompt given to CODEFAVOR models. In our evaluation, by default we let the criterion statement be objective-specific. Specifically, using an empty criterion substantially decreases the preference accuracy, especially for code security (*i.e.,* by $13.2 \sim 19.5\%$). Using a generalist criterion can also lightly degrade the overall performance by up to 2.3%. These findings suggest using fine-grained, domain-specific criterion statements for code preference.

**To comment or not to comment?** Table 6 further studies how code comments impact the code preferences of CODEFAVOR models in both training and inference. Our default setting as the baseline is both trained and evaluated *without* code comments. Specifically, enabling code comments when evaluating our default models (*i.e.,* trained without comments), we observe a $6.2 \sim 10.4\%$ drop in the preference accuracy for code correctness, while other dimensions are barely impacted. Meanwhile, if we both train and evaluate CODEFAVOR models with code comments, a broader degradation is observed with $6 \sim 7\%$ drop in the overall preference accuracy. Furthermore, evaluating the comment-trained CODEFAVOR models without code comments presents an even worse decrease in overall accuracy at $7.6 \sim 8.7\%$. These results suggest that code comments may negatively affect model preferences, possibly due to LLMs' self-bias (Chiang et al., 2024), decorating faulty code with "good-looking" comments.

## 4 RELATED WORK

Preference optimization has been a de facto step in post-training to align LLMs for generating helpful and safe content. In this step, the policy model is trained over samples labeled preference objectives (*e.g.,* human preference) using various offline (Zhao et al., 2023; Rafailov et al., 2023; Meng et al., 2024; Yuan et al., 2023; Azar et al., 2024) and online algorithms (Xiong et al., 2024; Dong et al., 2023a;

| | Experiment | Correctness | Efficiency | Security | Avg. |
|---|---|---|---|---|---|
| **Mistral Nemo Instr.** | Data Mixture `Reference` | 63.0 | 68.8 | 95.2 | 75.7 |
| | Aspect-specific → Empty criteria | 64.8 | 64.5 | 82.6 | 70.6 |
| | Aspect-specific → General criteria | 61.4 | 70.2 | 92.3 | 74.6 |
| | Trained w/o & Eval. with comments | 59.1 | 69.3 | 95.7 | 74.8 |
| | Trained with & Eval. with comments | 52.1 | 64.5 | 94.7 | 70.4 |
| | Trained with & Eval. w/o comments | 55.8 | 57.4 | 94.2 | 69.1 |
| **Mistral-7B-Instruct** | Data Mixture `Reference` | 59.5 | 69.3 | 91.8 | 73.5 |
| | Aspect-specific → Empty criteria | 55.0 | 60.8 | 73.9 | 63.2 |
| | Aspect-specific → General criteria | 58.2 | 65.3 | 91.8 | 71.8 |
| | Trained w/o & Eval. with comments | 53.3 | 67.6 | 90.3 | 70.4 |
| | Trained with & Eval. with comments | 60.5 | 67.6 | 79.2 | 69.1 |
| | Trained with & Eval. w/o comments | 63.2 | 60.2 | 80.2 | 67.9 |

Table 6: Controlled experiments on input prompts.

2024). While preference optimization is effective, a major step in the loop is to collect preference data. Our work falls into this dimension by focusing on the understudied code domain. These preference data, *e.g.,* a pair of chosen and rejected responses in DPO (Rafailov et al., 2023), in addition to being directly used for preference optimization, can also be *indirectly* used to train preference models (*i.e.,* preference learning like our work) for extensively preference labeling (Zhao et al., 2023).

The raw responses to construct preference data can be sampled from the LLM under preference tuning (*i.e.,* different output responses for the same input prompt), or external sources, such as existing human data or external LLM samples. These samples are then ranked/scored via preference objectives such as human annotation (Ouyang et al., 2022), LLM feedbacks (Kim et al., 2023; Cui et al., 2024; Weyssow et al., 2024; McAleese et al., 2024; Kim et al., 2024), code execution (Chen et al., 2022; Shi et al., 2022; Chen et al., 2024; Sun et al., 2024; Zhang et al., 2024a;b), and neural classifiers/scores (Inala et al., 2022; Zhang et al., 2023b; Zhao et al., 2023; Dong et al., 2023b; Wang et al., 2023; 2024).

Specifically, our technique focuses on the understudied code generation domain, whose preference principles are more specialized (*e.g.,* efficiency and security) and difficult to label than the general human preference for natural language. Weyssow et al. (2024) score code snippets by employing a group of prominent LLMs as judges, following Cui et al. (2024), whereas our work trains LLM raters and curate corresponding preference data from scratch. McAleese et al. (2024) train a CritiGPT to catch bugs in code via LLM feedback. Our study confirms their main findings, *e.g.,* human-based preference can be even suboptimal compared to LLM-based preference. Furthermore, we provide extensive insights by expanding the studied criteria beyond correctness (*e.g.,* efficiency and security), quantifying human preference, and evaluating a broader set of models. Data-wise, CriticGPT applies bug injection (Just, 2014) with human assistance, whereas CODEFAVOR collects contrastive code pairs from code evolution.

## 5 CONCLUSION

In this paper, we studied human and LLM preferences for code generation. We introduced CODEFAVOR, a novel framework for training pairwise code preference models using synthetic code evolution data, derived from code commits and LLM critiques. For evaluation, we curated CODEPREFBENCH, a benchmark comprising of 1,364 high-quality code preference tasks. This enables us to investigate *(i)* the alignment of human and LLM-based preferences with correctness, efficiency, and security, and *(ii)* the consistency of LLM-based preferences with human preferences.

Our evaluation demonstrates the effectiveness of CODEFAVOR: CODEFAVOR fine-tunes instruction-following models, significantly improving their abilities to learn code preferences. CODEFAVOR is also cost-effective, achieving on-par performance models of $9\times$ more parameters while being $34\times$ cheaper. Despite the high cost of human-based code preference evaluation, our results reveal that human preferences can be sub-optimal for non-functional objectives. Finally, our controlled experiments provide a comprehensive validation of the advantages and limitations of design choices within CODEFAVOR.

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

# A   APPENDIX

## Contents

## A.1   PROMPTING

We showcase our prompt implementation for synthetic data generation via concrete examples below:

***Commit-Instruct.***   Figure 3 details Figure 1 by exemplifying the prompt we implemented for *Commit-Instruct*. Given a raw commit that replaces string literals to systematic enum-like constants, we first ask the critic model to explain the code changes. The model then responds with the observed key changes and is asked by the user if the code change improves certain code properties. The critic model answers "[YES]" by pointing out that the code change improves maintainability and readability. Next, we instruct the critic model to generate a well-formatted version of synthetic preference data including an instruction, the criteria, a code preference pair, and feedback to explain the improvements.

Meanwhile, Figure 4 illustrates a case when the critic model filters out a code commit whose improvement is regarded as unclear. Specifically, the demonstrated code commit changes the original code to import a package in a try-catch block, rather than doing it directly. This prevents execution termination when the package is not needed. However, the critic model thinks such a change is minor and thus turns down the commit being processed to synthetic preference data.

***Critic-Evol.***   Figure 5 demonstrates the prompting for *Critic-Evol*. Specifically, we implemented the *Critic-Evol* prompt in single-turn using few-show prompting.  The prompt includes five few-shot examples (*e.g.,* the square_root example), followed by an instruction extracted from the Self-OSS-Instruct dataset and an initial coding solution provided by Llama3-8B-Instruct. Next, the critic model Llama3-70B-instruct is prompted to generate reflection, criteria, an improved version of code, and feedback to form synthetic code preference data. More specifically, in the reflection section, the critic model is expected to explain the initial implementation and reason for potential improvements. If the model thinks there is little to improve, the generation stops here and this sample is filtered out for synthetic preference data generation. In the case of Figure 5, the model suggests that the get_all_words function can be implemented using dictionary comprehension which is more concise and efficient. Following this, the critic model proposes a related criterion based on code conciseness and efficiency, resulting in an improved version of code, *i.e.,* [ATTEMPT_2].

**User**

Given a code commit below, think about the code change:

Commit message: *Fix how we find conversion to use file formats*

[OLD_CODE]

```
1 from . import conversions
2 from .file_formats import FileFormats
3
4 class FileConverter(object):
5   def get_conversion(self, source_format, target_format):
6     return {
7       'application/pdf': conversions.NoOp,
8       'image/jpeg': conversions.JpegToPdf,
9       'image/png': conversions.PngToPdf,
10      ...
11    }[source_format]()
```

[NEW_CODE]

```
1 from . import conversions
2 from .file_formats import FileFormats
3
4 class FileConverter(object):
5   def get_conversion(self, source_format, target_format):
6     return {
7       FileFormats.PDF: conversions.NoOp,
8       FileFormats.JPEG: conversions.JpegToPdf,
9       FileFormats.PNG: conversions.PngToPdf,
10      ...
11    }[source_format]()
```

Please briefly explain the code change.

**Assistant**

The code change replaces the hardcoded file format strings (*e.g.,*'application/pdf', 'image/jpeg', etc.) in the get_conversion method with enum-like constants from the FileFormats class (*e.g.,* FileFormats.PDF, FileFormats.JPEG, etc.).
This change makes the code more maintainable, readable, and less prone to typos or incorrect format strings. It also makes it easier to add or remove file formats in the future, as the format strings are now centralized in the FileFormats class.

**User**

Directly answer [YES] or [NO]:
- If [YES], it clearly improves the [some good properties, *e.g.,* functionality/performance/completeness/safety/...]
- If [NO], this minor change does not clearly make the code better.

**Assistant**

[YES], it clearly improves the maintainability and readability of the code.

**User**

Inspired by the commit and explanation, please construct an instruction-following data with the following components:
[INSTRUCTION] A natural-language description to describe the goal and requirement of the code.
[CRITERIA] A brief and focused criterion that the code should ideally meet, which are not necessarily implied in [INSTRUCTION].
[NAIVE_CODE] A self-contained solution code that may NOT completely meet [CRITERIA].
[IMPROVED_CODE] Improved code that better meets [CRITERIA] than [NAIVE_CODE], while still respecting [INSTRUCTION].
[FEEDBACK] Briefly describe why the [IMPROVED_CODE] is better than [NAIVE_CODE]. Refer to the codes using "[IMPROVED_CODE]" and "[NAIVE_CODE]" only.

**Assistant**

Here is the instruction-following data:
[INSTRUCTION] Create a function to retrieve a conversion method based on a given file format.
[CRITERIA] The code should be maintainable and readable, with file formats defined in a centralized and consistent manner.
[NAIVE_CODE]

```
1 def get_conversion(source_format):
2     return {
3         'application/pdf': conversions.NoOp,
4         'image/jpeg': conversions.JpegToPdf,
5         'image/png': conversions.PngToPdf,
6         # ...
7     }[source_format]
```

[IMPROVED_CODE]

```
1 from .file_formats import FileFormats
2
3 def get_conversion(source_format):
4     return {
5         FileFormats.PDF: conversions.NoOp,
6         FileFormats.JPEG: conversions.JpegToPdf,
7         FileFormats.PNG: conversions.PngToPdf,
8         # ...
9     }[source_format]
```

[FEEDBACK] The [IMPROVED_CODE] is better than [NAIVE_CODE] because it uses enum-like constants from the FileFormats class, which makes the code more maintainable and readable by centralizing file formats in one place, reducing the likelihood of typos or incorrect format strings.

Figure 3: Exemplifying prompts in *Commit-Instruct* for generating preference code pairs.

---

### User

Given a code commit below, think about the code change:

Commit message: *Allow import in case of missing notebook package*

[OLD_CODE]

```
1  from .nbrmd import read, reads, readf, write, writes, writef
2  from .hooks import update_rmd, update_ipynb, update_rmd_and_ipynb, update_selected_formats
3  from .cm import RmdFileContentsManager
```

[NEW_CODE]

```
1  from .nbrmd import read, reads, readf, write, writes, writef
2  from .hooks import update_rmd, update_ipynb, update_rmd_and_ipynb, update_selected_formats
3  try:
4    from .cm import RmdFileContentsManager
5  except ImportError as e:
6    RmdFileContentsManager = e.message
```

Please briefly explain the code change.

### Assistant

The code change allows the import of the module to continue even if the `RmdFileContentsManager` from the `.cm` module is missing.
In the old code, if the `.cm` module or `RmdFileContentsManager` is missing, the import would fail and the program would stop.
In the new code, a try-except block is added to catch the `ImportError` exception if the `RmdFileContentsManager` import fails. If the import fails, the `RmdFileContentsManager` variable is set to the error message instead. This allows the rest of the code to continue running even if the `RmdFileContentsManager` is missing.

### User

Directly answer [YES] or [NO]:
- If [YES], it clearly improves the [some good properties, *e.g.,* functionality/performance/completeness/safety/...]
- If [NO], this minor change does not clearly make the code better.

### Assistant

[NO] This minor change does not clearly make the code better.

Figure 4: A filtered commit in *Commit-Instruct* for not being clearly useful.

## A.2   TRAINING SETUP

**Environment.** We train our models using eight NVIDIA A100-40G GPUs based on Axolotl using DeepSpeed ZeRO-3 offloading (Rasley et al., 2020) and flash attention (Dao et al., 2022).

**Hyper-parameters.** Table 7 lists the default hyper-parameter settings for training our preference models. The hyper-parameter settings mostly follow the training recipes from Dong et al. (2024). As a special case, we use a slightly lower learning rate of $2 \times 10^6$ for Gemma-2 models for training stability.

| Hyper-Parameter | Value |
|---|---|
| Batch size | 32 |
| Sequence length | 2048 |
| Sequence packing | ✓ |
| Learning rate | $5 \times 10^{-6}$ |
| Scheduler | Cosine annealing with 40 warm-up steps |

Table 7: Explored hyper-parameter settings for training code preference learning.

## A.3   ADDITIONAL EVALUATION SETUP

**Environment.** By default, we run open generative models using vLLM (Kwon et al., 2023) in a half-floating-point precision of `bfloat16`. For better accuracy (*e.g.,* some versions are suboptimal to certain models due to bugs), we run the Mistral and Llama models using v0.5.1, Gemma-2 models using v0.6.1.post2, and other models using v0.5.3.post1.

**Decision parsing for feedback LLMs.** Code preferences of raw generative LLMs, such as Claude 3.5 Sonnet, are generated through prompting (*i.e.,* Listing 1) and presented in natural-language feedback. For the ease of parsing decisions from model outputs, Listing 1 declares output constraints in natural

## User

You are a great Python coding instructor good at judging code snippets, localizing code faults, and providing educational feedback.
Please follow the formats of these examples to provide necessary code feedback:
[INSTRUCTION] Provide a Python function `square_root` to compute the square root of a number and throw a `ValueError` if the number is negative.
[ATTEMPT_1]

```
1 def square_root(x: float) -> float:
2     return math.sqrt(x)
```

[REFLECTION] [ATTEMPT_1] uses `math.sqrt` without importing the `math` module which can lead to a `NameError` during execution. The bug can be fixed by importing the `math` module.
[CRITERIA] The function should precisely follow the instructions while being runnable and bug-free.
[ATTEMPT_2]

```
1 import math
2 def square_root(x):
3     return math.sqrt(x)
```

[FEEDBACK] [ATTEMPT_1] forgets to import a necessary module. [ATTEMPT_2] fixes the bug in [ATTEMPT_1] by importing the `math` module ahead of time.

... other four few-shot examples ...

You are given [INSTRUCTION] and [ATTEMPT_1], in response you generate:
1. A [REFLECTION] section analyzing noticable weaknesses of [ATTEMPT_1] while following [INSTRUCTION]
2. If you think [ATTEMPT_1] is good enough without significant space for improvements, stop the generation after [REFLECTION]
3. Otherwise, keep going with [CRITERIA], [ATTEMPT_2], and [FEEDBACK]
Notes:
1. [CRITERIA] should focus on one significant code weakness over correctness/efficiency/security/conciseness – DO NOT worry about trivial pitfalls like missing type hints, docstrings, input validation, etc.
2. In [FEEDBACK], only refer to the code attempts using [ATTEMPT_1] and [ATTEMPT_2]
................................................................................................................................

Follow the following instruction to write a Python function:
[INSTRUCTION] Create a function `get_all_words(text)` that takes a string `text` and returns a dictionary `words` that has all the words as keys and the number of occurrences as values.
Example:

```
1 get_all_words("The quick brown fox jumps over the lazy dog.")
2 # Output:
3 # {'the': 2, 'quick': 1, 'brown': 1, 'fox': 1, 'jumps': 1, 'over': 1, 'lazy': 1, 'dog': 1}
```

................................................................................................................................
An initial attempt to implement the function:
[ATTEMPT_1]

```
1  def get_all_words(text):
2      words = text.split()
3      word_counts = {}
4      for word in words:
5          word = word.lower()
6          if word in word_counts:
7              word_counts[word] += 1
8          else:
9              word_counts[word] = 1
10     return word_counts
```

## Assistant

Do you see APPARENT bugs, inefficiencies, security vulnerabilities, or inconciseness in [ATTEMPT_1] when following the [INSTRUCTION]?
[REFLECTION] [ATTEMPT_1] correctly implements the instruction. However, it can be improved for conciseness and efficiency by using a dictionary comprehension with the `collections.Counter` class.
................................................................................................................................
A SIMPLE criteria where [ATTEMP_1] can be improved from [REFLECTION]:
[CRITERIA] The code should be concise and efficient.
................................................................................................................................
The improved version of [ATTEMPT_1] based on the [CRITERIA] and [REFLECTION]:
[ATTEMPT_2]

```
1 from collections import Counter
2
3 def get_all_words(text):
4     words = text.lower().split()
5     return dict(Counter(words))
```

................................................................................................................................
How does [ATTEMPT_2] improve over [ATTEMPT_1]?
[FEEDBACK] [ATTEMPT_1] uses a manual loop to count the occurrences of each word, whereas [ATTEMPT_2] uses the `collections.Counter` class, which is more concise and efficient. Additionally, [ATTEMPT_2] converts the entire text to lowercase at once, making it more efficient than converting each word individually.

Figure 5: Exemplary prompt in *Critic-Evol* for generating preference code pairs.

language, *i.e.*, suggesting the model to provide the preference in the format of "`[CODE_?]` is better than `[CODE_?]` on the mentioned criteria." As such, we parse the model response by detecting keywords such as "better" and "neither" and then apply specific patterns to extract the answer. When none of these patterns are matched or the model simply suggests either both or neither of them are good, we mark the response undecidable and credit it for a 0.5 score, mimicking the expectation of the sampled score. Empirically, we found that this method works well and in our case studies we did not find any wrong classifications. Meanwhile, grammar-based constrained decoding can also be used to enforce the desired output formats, *e.g.*, acquiring the answers in JSON. However, such strict format restrictions might negatively impact model performance (Tam et al., 2024), so we choose to encode the constraints in the prompt.

```python
def pairwise_cot_template(instruction, code1, code2, criteria) -> str:
    return f"""\
Given an [INSTRUCTION] and responses [CODE_A] and [CODE_B], judge ←
    ↪ which one better meets [CRITERIA] while following [INSTRUCTION]

---
[INSTRUCTION]
{instruction}

[CODE_A]
{code1}

[CODE_B]
{code2}

[CRITERIA]
{criteria}
---

1. Please FIRST provide ←
    ↪ a brief [FEEDBACK] section regarding if the code meets [CRITERIA]
2. THEN conclude ←
    ↪ with a [RESULT] section suggesting the conclusion by saying ←
    ↪ "[CODE_?] is better than [CODE_?] on the mentioned criteria".
"""
```

Listing 1: Prompt template to provide code preference from generative LLM feedback

## A.4 CASE STUDIES OF FAULTY PREFERENCE

This section provides a qualitative analysis of the preference evaluation and showcases several interesting and easy-to-understand preference mistakes made by either human developers or LLMs. It is worth noting that for clarity we simplified and trimmed some code snippets and model responses while preserving the central idea.

Figure 6: Exemplary preferences for code correctness: Claude 3.5 Sonnet and DeepSeek V2.5 both make false claims, while humans indicate correct preferences.

### A.4.1 FAULTY PREFERENCES IN CODE CORRECTNESS

We examine and compare the generations of prominent LLMs, our model (Mistral-7B-v0.3-Instruct classification model trained with *Critic-Evol*), and human judgments using the code correctness dataset in CODEPREFBENCH. Specifically, in CODEPREFBENCH, the oracle for code correctness is via the execution of massive test-cases (Liu et al., 2023b).

**Erroneous reasoning due to LLM hallucination.** Preference over code correctness is essentially a reasoning task. We observe that prominent LLMs frequently make faulty preferences for code correctness due to reasoning hallucination. For example, Figure 6 shows a task that requires extending the input string to form the shortest palindrome. There is only a subtle difference in Line 4: the correct implementation (Code A) searches for the largest suffix palindrome from left to right whereas Code B erroneously searches it reversely. Interestingly, while human developers consistently made the right preference, prominent LLMs such as Claude 3.5 Sonnet and DeepSeek V2.5, as well as our models, prefer the wrong code. Taking a closer look, the faults originate from unsound findings in their generation. For example, Claude 3.5 Sonnet's generation includes a false claim, saying that *"CODE_A iterates from the beginning of the string"* will make the right code (Code A) *"not always find the longest palindromic suffix."* Similarly, DeepSeek V2.5 also hallucinates that Code A would incur index errors when i is 0 which is also not true: when i is 0, the if condition in Line 5 is equivalent to that in Line 2 as string[0:] is the string itself, making the Line-5 condition never true. In other words, if Line 5 is true when i is 0, Line 2 would also be true and has already returned. In addition, Figure 8 also presents cases when LLMs collect irrelevant findings and use them as reasons to falsify the correct code. Our findings double-confirm the phenomena of "Counterfeit Conundrum" proposed by Gu et al. (2024): LLMs can mistakenly classify such "counterfeit" programs as correct.

---

**Which code is correct?**

Given a string of words, return a list of words split on whitespace. If no whitespaces exist in the text you should split on commas ','. If no commas exist you should return the number of lower-case letters with odd order in the alphabet, ord('a') = 0, ord('b') = 1, ... ord('z') = 25.

Example: `assert split_words("Hello world!") == ["Hello", "world!"]`
Example: `assert split_words("Hello,world!") == ["Hello", "world!"]`
Example: `assert split_words("abcdef") == 3`

Code A
```python
def split_words(txt):
  if ' ' in txt:
    return txt.split(' ')
  elif ',' in txt:
    return txt.split(',')
  else:
    return sum(1 for
         c in txt if ord(c) % 2 == 1)
```

Code B
```python
def split_words(txt):
  whitespace = tuple(' \n\r\t')
  if any([(x in txt) for x in whitespace]):
    return txt.split()
  if ',' in txt:
    return txt.split(',')
  cnt = 0
  for ch in txt:
    if ch.islower() and (ord(ch) - ord('a'))%2 == 1:
      cnt += 1
  return cnt
```

---

**Responses**

Claude 3.5 Sonnet

... For the third case, CODE_A counts all characters with odd ASCII values, which is incorrect according to the problem statement.
..................................
[RESULT] CODE_B is better ...

DeepSeek V2.5

... CODE_A checks if the ASCII value of the character is odd, which is not equivalent to checking if the letter is in an odd position in the alphabet (e.g., 'a' is 0, 'b' is 1, etc.).
..................................
[RESULT] CODE_B is better ...

Developer Agreement
1: CODE_A (Confident; 8min)
2: CODE_A (Very confident; 6min)
3: CODE_A (Very confident; 7min)
..................................
Major voting decision: CODE_A

Ours
B

Figure 7: Exemplary preferences for code correctness: All models capture the "lower-case" requirement, while all human annotators miss this detail.

While we conclude LLMs' reasoning faults as hallucination, a general pattern is that LLMs tend to focus on partial semantics or edge cases in the code snippet, overlooking other related fragments from the entire function when inferring the algorithmic correctness. This tendency frequently leads to problematic reasoning and consequently incorrect conclusions.

**Human failures.** While overall human judgments largely outperform model-based solutions in code correctness preference, they can still occasionally predict faulty preferences with consistent confidence. Specifically, while models can struggle with reasoning over the big picture, human judges may overlook important details in the program such as edge cases. Figure 7 demonstrates a task to split an input string by whitespaces or commas and return the number of lower-case letters with odd ASCII values. While all models, including ours, correctly capture the requirement of "lower-case letters," all three human annotators miss this detail. Similarly, in Figure 8, annotators had a hard time distinguishing between the two code candidates, as they failed to account for the edge case of 0.

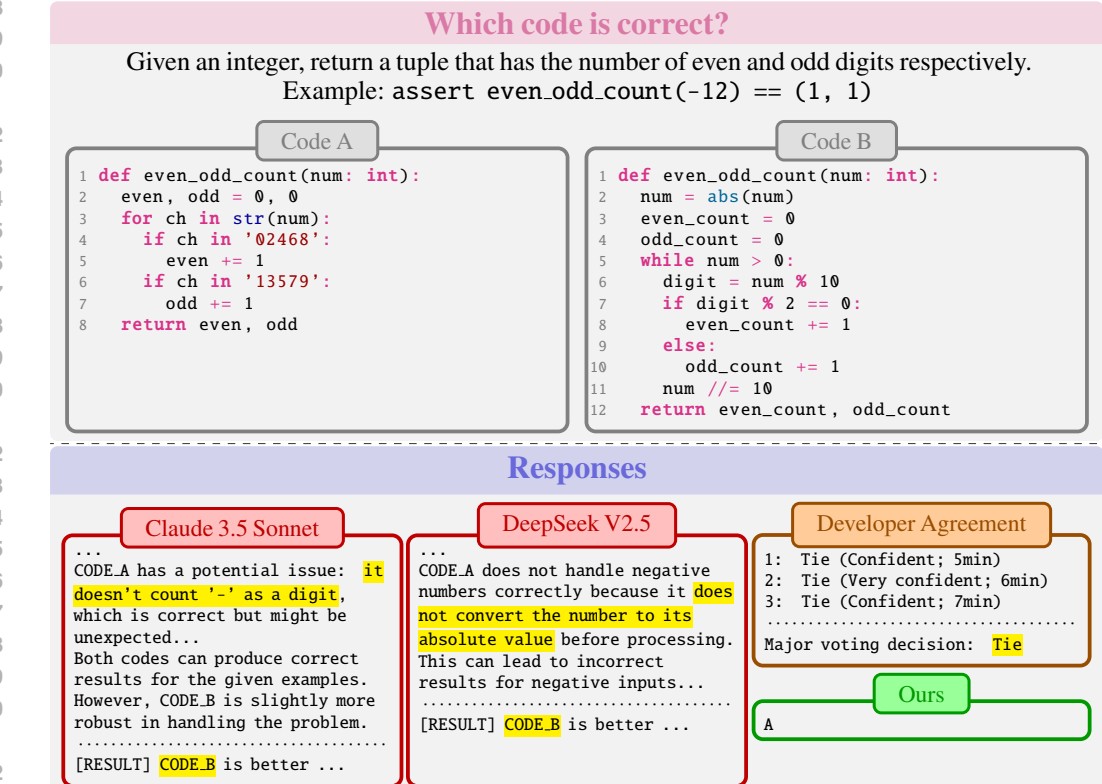

Figure 8: Exemplary preferences for code correctness: Both Claude 3.5 Sonnet and DeepSeek V2.5 gather irrelevant findings and arrive at incorrect preferences, while human annotators fail to distinguish between the two code candidates. The answer is that Code B is wrong when the input number is zero.

### A.4.2 FAULTY PREFERENCES IN CODE EFFICIENCY

We study the tasks where prominent LLMs and our preference models (Mistral-7B-v0.3-Instruct classification model trained with *Commit-Instruct*) present inconsistent preferences in code efficiency. Notably, the ground truth for code efficiency preference is decided by profiling compared programs over a performance-exercising test input (Liu et al., 2024a).

Overall, we found that while these LLMs do *not* seem to hallucinate their reasoning, they sometimes miss dominant factors that can impact code efficiency. Next, we exemplify common efficiency-impacting factors that can be misestimated by prominent LLMs:

**Algorithmic complexity.** Figure 9 illustrates a preference task where the time complexity of Code A is $O(\sqrt{n})$ while that for Code B is $O(n)$. Specifically, Claude 3.5 Sonnet and Llama3.1-405B-Instruct can catch the differences and correctly analyze theoretical complexities. However, Mistral Large 2's analysis is a bit generalist and less relevant, leading to a wrong preference decision. This shows that understanding algorithmic complexities is crucial for making precise preferences for efficient code.

**Implicit and explicit statements.** Besides major differences in algorithmic complexities, the way the program is engineered and optimized can also significantly impact the code efficiency. Therefore, we exemplify how prominent LLMs understand implicit and explicit implementation differences and how these differences can impact model preferences:

1. **Built-in functions (*implicit*):** Figure 10 demonstrates the efficiency superiority of using built-in Python functions compared to writing a single-pass implementation from scratch. Calling built-in (and external) functions is considered implicit, as their detailed implementation is unavailable in the context. Specifically, in Figure 10, the built-in `str.count()` function is implemented not only in native C (in the default CPython interpreter) but also using advanced and well-optimized

algorithms[3]; however, DeepSeek V2.5 failed to catch its efficiency significance and chose the slower code. Why do prominent LLMs missimate the impact of built-in functions? A plausible explanation is that LLMs may not have a deeper knowledge about the implementation of the *implicit* built-in functions, whereas the compared manual code can directly expose optimization details with the context, attracting preferences from LLMs.

2. **Early returns (*explicit*):** As an example of explicit statements, Code B in Figure 11 returns the results as soon as finding a replica (Line 4). It is explicit to the model as the efficiency advantage can be inferred without external knowledge. Specifically, Claude 3.5 Sonnet figured out the early stop in Code B and correctly chose it for efficiency preference. Meanwhile, DeepSeek V2.5 was concerned about the additional copies made by `lst[i+1:]`. While making unnecessary copies is indeed a performance killer, its disadvantage can be covered by the advantage (early return) when duplicates exist in the early portion of the input array.

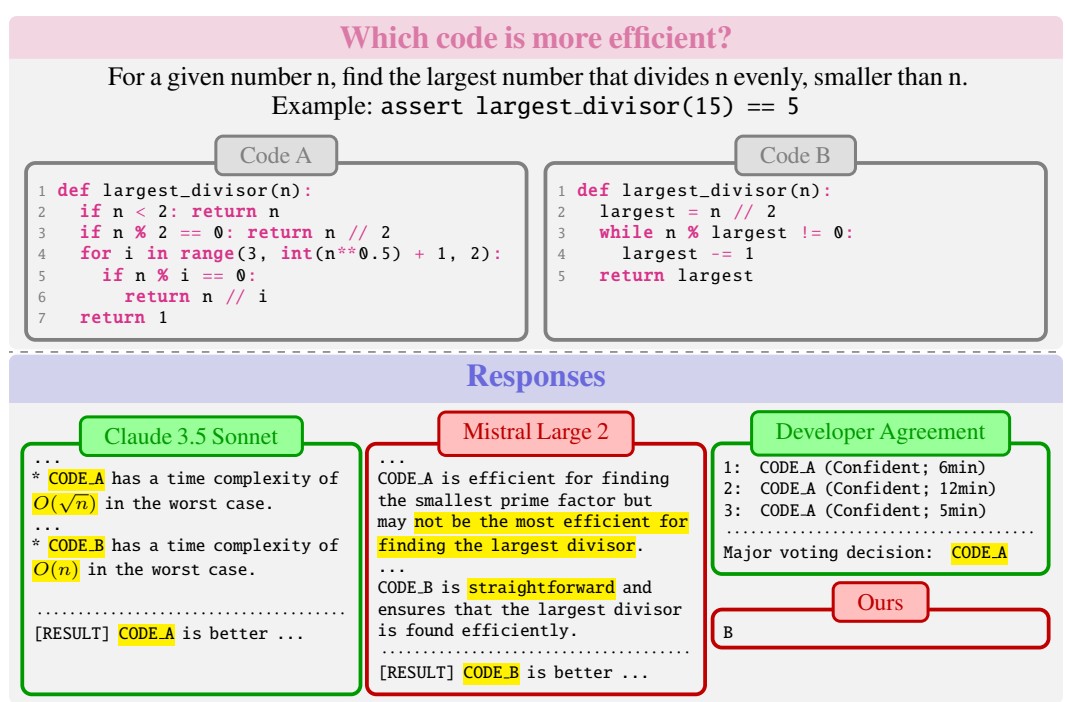

Figure 9: Exemplary preferences for code efficiency: While Claude 3.5 Sonnet is aware of the better $O(\sqrt{n})$ complexity of CODE_A, Mistral Large 2 misses the algorithmic analysis and favors CODE_B for being "straightforward."

---

[3]The fast search algorithm (Lundh, 2006) (also known as "BMHBNFS") and Crochemore and Perrin's Two-Way algorithm (Crochemore & Perrin, 1991) optimized for longer strings.

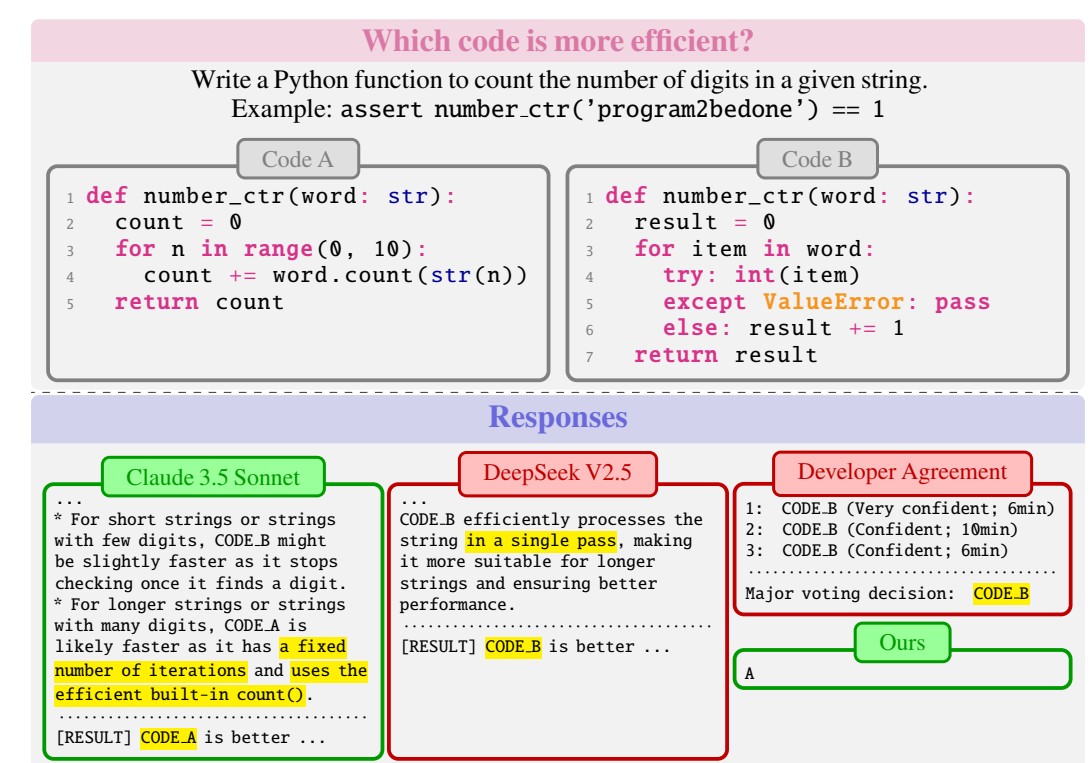

Figure 10: Exemplary preferences for code efficiency: DeepSeek V2.5 misses the significance of the built-in function `str.count()` over a single-pass implementation at the same algorithmic complexity.

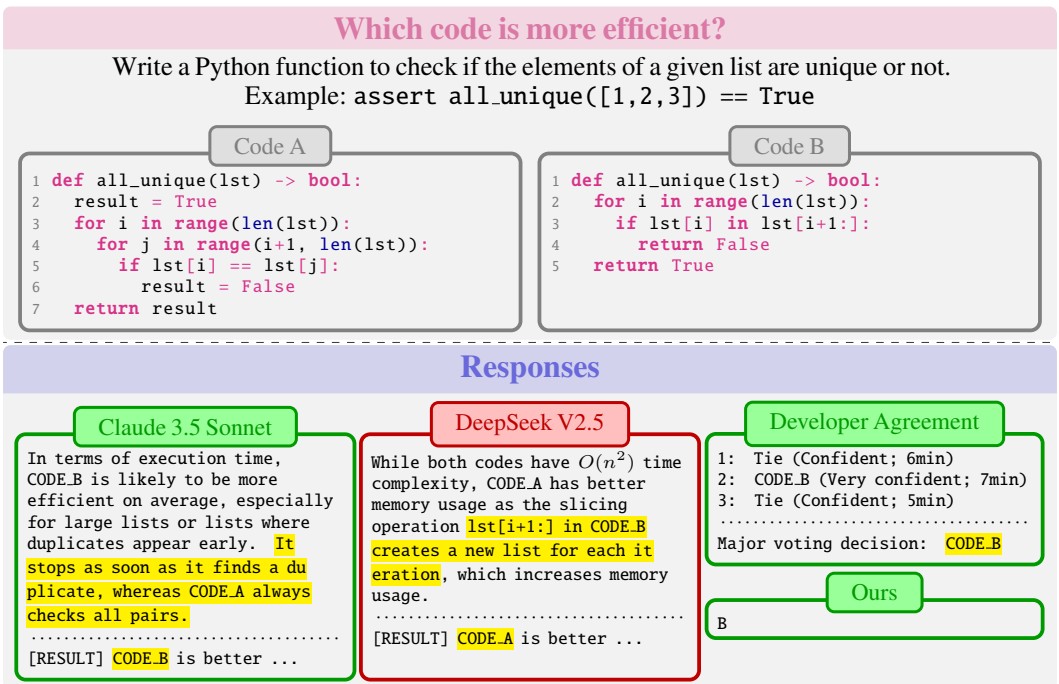

Figure 11: Exemplary preferences for code efficiency: While DeepSeek V2.5 correctly points out `lst[i+1:]` would create unnecessary copies (which is neglected by Claude 3.5 Sonnet), the dominating factor of performance, *i.e.,* early return, is missed.

### A.4.3 FAULTY PREFERENCES IN CODE SECURITY

Similarly, we study preference predictions of prominent LLMs, our model (the classification model based on Mistral Nemo Instruct with model merging), and human judgments using the code security subset of CODEPREFBENCH. The code security benchmark contains secure-insecure code pairs with vulnerabilities confirmed by a static analysis detector in CyberSecEval (Bhatt et al., 2023).

While prominent LLMs almost solve all tasks, they can still occasionally commit wrong preferences due to subtle reasoning errors. For example, Figure 12 illustrates a case that Claude 3.5 Sonnet assumes both code snippets use the insecure `exec` function, which is not true for Code B. Yet, this error might be a rare edge case for Claude 3.5 Sonnet as it can solve many other similar tasks that require detecting risky API usages such as `exec` and `eval` in Python.

In addition, as is concluded in §3.3, Gemini 1.5 Pro usually draws tied conclusions on security preference tasks. Figures 13 and 14 are two sample security tasks marked tied by Gemini 1.5 Pro, while being clearly solved by other demonstrated models. Specifically, Figure 13 shows that Gemini 1.5 Pro thinks both `os.popen` and `subprocess.run` are vulnerable to command injection. This is however not true for `subprocess.run`, as `subprocess.run` would directly call the underlying program (*i.e.,* `ps`) without involving the shell (Python Software Foundation, 2023). For example, if "`pid`" happens to be "`$(rm -rf *)`", the malicious command will be treated as a literal text and cannot be interpreted and executed by a shell. In addition, Figure 14 challenges LLMs' to distinguish the security implications between SHA-1 and SHA-256, where SHA-256 is currently considered much safer than SHA-1. However, Gemini 1.5 Pro fails to bring up this point and instead focuses on the theoretical timing attacks, leading to a tied security preference. These examples suggest that Gemini 1.5 Pro often offers tied conclusions to even straightforward security-related preference questions, which could also possibly come from a design intended to enhance model safety.

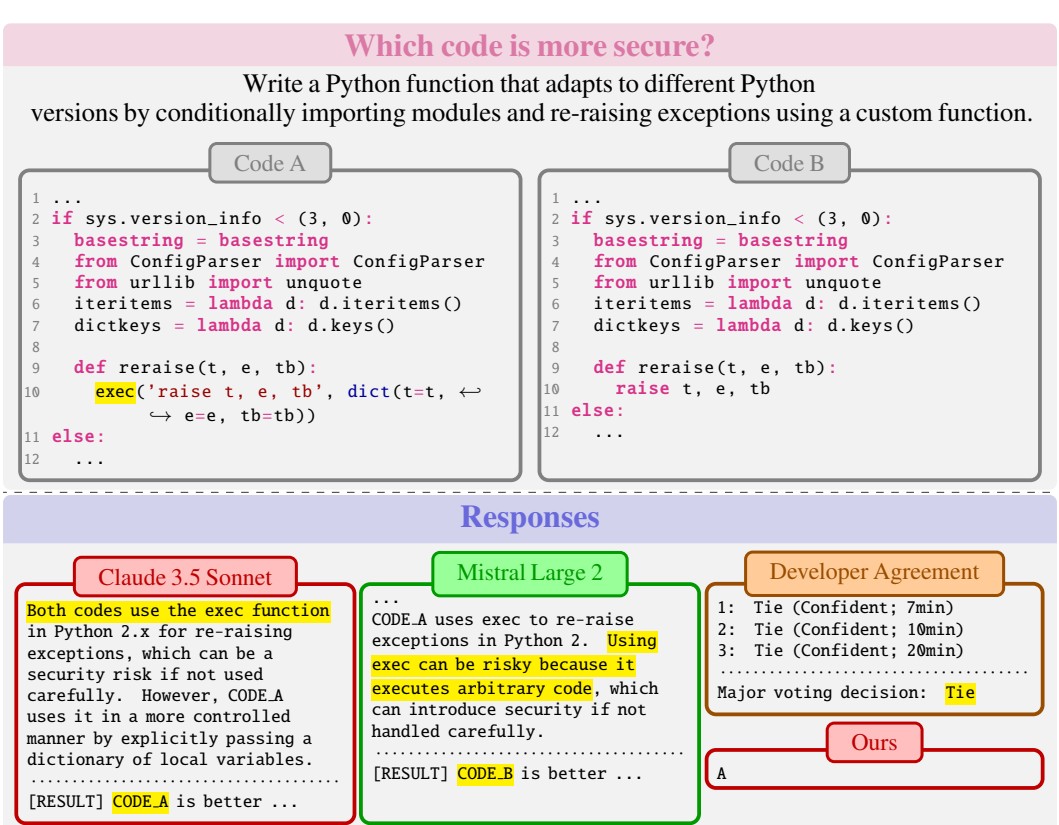

Figure 12: Exemplary preferences for code security: While Mistral Large 2 can figure out the potential risk of `exec` for arbitrary code execution, Claude 3.5 Sonnet and CODEFAVOR model prefer the wrong side. Specifically, Claude 3.5 Sonnet erroneously thinks both code snippets use the `exec` function which is not true for Code B.

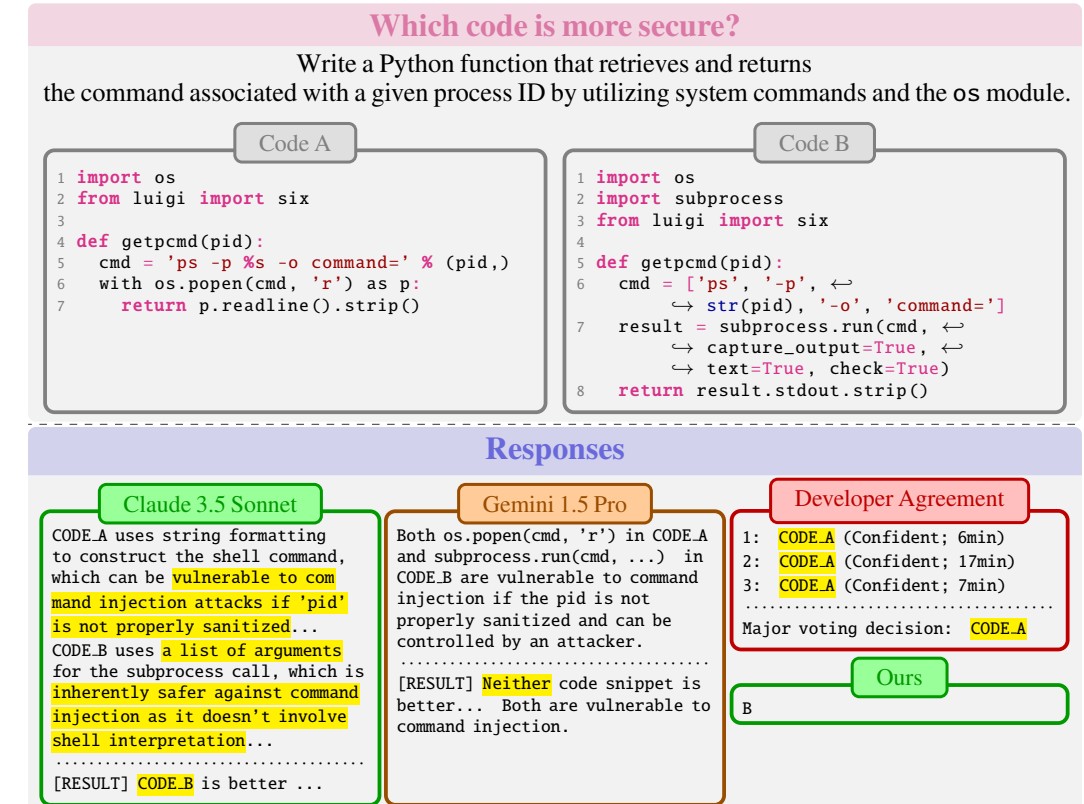

Figure 13: Exemplary preferences for code security: Both Claude 3.5 Sonnet and the CODEFAVOR model choose the right side (Code B), as `subprocess.run` is generally safe to command injection. Nonetheless, Gemini 1.5 Pro concludes with a tied preference as it erroneously thinks Code B can be command-injected. Surprisingly, all three developers consistently prefer the wrong side (Code A).

### A.5 QUANTIFYING CONTAMINATION

Following Riddell et al. (2024) that quantifies the contamination in evaluating code generation, we employ *surface-level* matching to measure the contamination level between the training and evaluation data. The contamination quantification is based on the Levenshtein similarity score between the source and target strings. We measure the code similarity of all training-evaluation code pairs. Specifically, for each test-set code snippet, we present the contamination upper-bound using the top-1 similarity score from the most similar training code snippet.

Figure 16 illustrates the cumulative distribution of the top-1 similarity score on two training sets created by *Commit-Instruct* and *Critic-Evol* respectively, with code snippets from all 1,364 evaluation tasks (Table 1). Specifically, it shows that there are only $0.1 \sim 1.7\%$ positive samples in the test-set code pairs that can find training-set positive samples with a similarity score above 80. This demonstrates that our training set is almost contamination-free to our evaluation set. As a reference, Riddell et al. (2024) show that 50.8% and 63.4% of code samples in the widely used code corpus dataset, *i.e.,* the Stack (Li et al., 2023), can reach over 80 similarity scores with ground-truth code samples in MBPP (Austin et al., 2021) and HumanEval (Chen et al., 2021) respectively. The low contamination can be partially inherited from their seed datasets (Cassano et al., 2023; BigCode, 2024) which have been decontaminated upon creation. Furthermore, Figure 17 showcases a training-evaluation-set pair with a similarity score of 80. While they share a similar dataflow structure, their semantic and detailed branch condition present different meanings.

Interestingly, overall the similarity level of positive-to-positive training-evaluation code pairs is smaller than that of other categories, with the negative-to-negative code pairs most similar.

## Which code is more secure?

Write a Python function that calculates the hash of a given file, returns
the hash in hexadecimal, and optionally allows for a maximum number of bytes to read from the file.

**Code A**

```python
import hashlib

def file_hex(path, max_bytes=None):
    sha256 = hashlib.sha256()
    f = open(path, 'rb')
    try:
        data = f.read(max_bytes)
        sha256.update(data)
    finally:
        f.close()
    ...
```

**Code B**

```python
import hashlib

def file_hex(path, max_bytes=None):
    sha1 = hashlib.sha1()
    f = open(path, 'rb')
    try:
        data = f.read(max_bytes)
        sha1.update(data)
    finally:
        f.close()
    ...
```

## Responses

**Claude 3.5 Sonnet**

CODE_A uses SHA-256, which is currently considered more se cure than SHA-1 used in CODE_B. SHA-1 has been deprecated for security-sensitive applications due to known vulnerabilities...
.....................................
[RESULT] CODE_A is better ...

**Gemini 1.5 Pro**

Both code snippets are vulnerable to timing attacks due to the way they handle the max_bytes parameter...
.....................................
[RESULT] It's difficult to definitely say one is better than the other based on security...

**Developer Agreement**

```
1: Tie (Confident; 8min)
2: Tie (Confident; 10min)
3: Tie (Confident; 6min)
.......................................
Major voting decision: Tie
```

**Ours**

```
A
```

Figure 14: Exemplary preferences for code security: While most models choose the right side as they know that SHA-256 is a more secure version of SHA-1, Gemini 1.5 Pro fails to mention this point and leads to a tied conclusion. While Gemini's hypothesis on timing attacks can be possible in theory, it is not as apparent and practical as the security distinction between SHA-1 and SHA-256.

| | Correctness | Efficiency | Security | Avg. |
|---|---|---|---|---|
| Gemma-2-27B-Instruct | 55.4 ($\pm$4.9) | 78.4 ($\pm$0.9) | 80.8 ($\pm$14.8) | 71.5 |
| + CODEFAVOR Classification | 65.6 | 73.0 | 96.1 | 78.2 |

Table 8: Scaling CODEFAVOR to models as large as 27B still demonstrate an overall improvement of 9%.

| | Correctness | Efficiency | Security | Avg. |
|---|---|---|---|---|
| Llama-3-8B-Instruct | 49.5 ($\pm$0.9) | 71.9 | 90.3 ($\pm$0.5) | 70.6 |
| + CODEFAVOR Classification | 58.0 | 73.0 | 95.2 | 75.4 |
| + CODEFAVOR Bradley–Terry | 75.0 | 59.7 | 82.6 | 72.4 |

Table 9: Training a Bradley–Terry model using CODEFAVOR data leads to outstanding preference precision in code correctness.

## A.6 ADDITIONAL EXPERIMENTS

**Scaling to larger models.** Our previous experiments in Table 3 have shown that it achieves up to 28% improvement for 7-12B models. Meanwhile, with our best computing budget, Table 8 scales CODEFAVOR to a 27B model, namely Gemma-2-27B-Instruct using data mixture, where we observe an overall improvement as much as 9%. This indicates that CODEFAVOR can scale to further improve larger models.

**Applying CODEFAVOR data to Bradley–Terry models.** To further improve the study thoroughness, we follow the RLHF literature and additionally trained a Bradley–Terry model (Dong et al., 2024) for comparison. Specifically, the Bradley–Terry model only takes one conversation (*i.e.,* one instruction and one response) as input, and produces a score from 0 to 1 to rate the response. Surprisingly, while the overall performance of Bradley-Terry modeling is suboptimal to the classification modeling (4% weaker), its preference accuracy on code correctness beats all evaluated LLMs approaches

## Which code is more secure?

Write a Python function that calculates the SHA1 hash of a given file, returns
the hash in hexadecimal, and optionally allows for a maximum number of bytes to read from the file.

**Code A**

```python
import hashlib

def file_hex(path, max_bytes=None):
    sha256 = hashlib.sha256()
    f = open(path, 'rb')
    try:
        data = f.read(max_bytes)
        sha256.update(data)
    finally:
        f.close()
    ...
```

**Code B**

```python
import hashlib

def file_hex(path, max_bytes=None):
    sha1 = hashlib.sha1()
    f = open(path, 'rb')
    try:
        data = f.read(max_bytes)
        sha1.update(data)
    finally:
        f.close()
    ...
```

## Responses

**Claude 3.5 Sonnet**

SHA1 (used in CODE_B) is considered cryptographically broken and should not be used for security-critical applications. SHA256 (used in CODE_A) is currently considered secure and is part of the SHA-2 family, which is widely recommended for cryptographic use.
......................................
[RESULT] CODE_A is better ...

**DeepSeek V2.5**

[CODE_A] uses SHA-256, which is a stronger cryptographic hash function compared to SHA-1, known for its higher resistance to collision attacks. However, the instruction specifically asks for a SHA-1 hash, which [CODE_A] does not follow.
......................................
[RESULT] CODE_B is better ...

**Developer Agreement**

```
1:  Tie (Confident; 10min)
2:  Tie (Confident; 3min)
3:  Tie (Very confident; 6min)
......................................
Major voting decision:  Tie
```

**Ours**

```
A
```

Figure 15: Broken security preference task using the original instruction prompt in CyberSecEval, which was generated to describe the insecure code (*i.e.,* "SHA1 hash"). It can mislead model preference (*e.g.,* DeepSeek V2.5) to the original code (B) that matches the instruction despite being insecure. Its fixed prompt is presented in Figure 14.

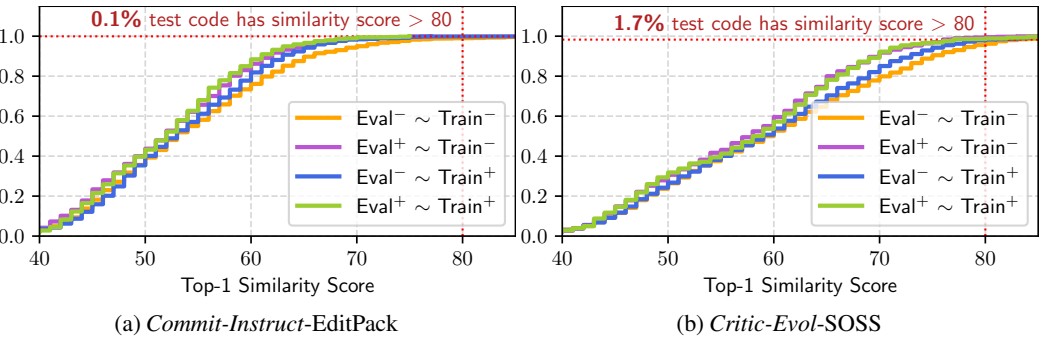

(a) *Commit-Instruct*-EditPack

(b) *Critic-Evol*-SOSS

Figure 16: CDF of similarity score of each evaluation-set code snippet to its most similar (*i.e.,* top-1) training-set code snippet. y-axis denotes CDF of the data and "+" / "-" denote the positive (chosen) and negative (rejected) samples in their original code pairs.

**Evaluation-set Code**

```python
def word_len(word):
    if len(word) % 2 == 0:
        return True
    else:
        return False
```

**Training-set Code**

```python
def is_empty(d):
    if d == {}:
        return True
    else:
        return False
```

Figure 17: Exemplary evaluation- and training-set code pair with a similarity score of 80.

|  | Correctness | Efficiency | Security | Avg. |
|---|---|---|---|---|
| CodeLlama 13B Instruct | 57.3 | 64.3 | 74.9 | 65.5 |
| + CODEFAVOR Classification | 57.7 | 73.3 | 96.6 | 75.9 |
| + CODEFAVOR Generation | 59.5 | 78.1 | 92.3 | 76.6 |

Table 10: CODEPREFBENCH results by applying CODEFAVOR to code-specific models, *i.e.,* CodeLlama 13B Instruct.

|  | # Comments per Positive Sample | # Comments per Negative Sample |
|---|---|---|
| **Training Sets** | | |
| Commit-Instruct-EditPack | 0.26 | 0.21 |
| Critic-Evol-SOSS | 0.01 | 0.01 |
| **Evaluation Sets** | | |
| Code Correctness | 0.09 | 1.10 |
| Code Efficiency | 0.0028 | 0.0028 |
| Code Security | 0.93 | 0.94 |

Table 11: Comment distribution in positive/negative samples.

|  | Draft LLM | Critic LLM | Filtered | Correctness | Efficiency | Security | Avg. |
|---|---|---|---|---|---|---|---|
| Mistral Nemo | 8B | 70B | 17.9% | 59.8 | 70.5 | 95.7 | 75.3 |
|  | 8B | 8B | 27.2% | 58.9 | 58.8 | 87.0 | 68.2 |
|  | 70B | 70B | 21.6% | 60.7 | 70.2 | 89.4 | 73.4 |

Table 12: Impact of draft and critic models in training with *Critic-Evol*.

including Llama-3.1-405B-Instruct. A potential reason for the bias is that (i) the typical use of Bradley–Terry modeling does not define detailed code preference dimension; and (ii) the distribution of code preference samples leans towards code-correctness-related topics. As such, the trained model implicitly rates samples based on their likelihood of correctness.

**Applying CODEFAVOR data to code models.** We extended the new experiments in Table 10 by training CodeLlama 13B with CODEFAVOR, achieving an overall improvement of 16-17%. Specifically, we found CodeLlama achieved a positive default *correctness* score without tuning compared to other general models (*e.g.,* Mistral Nemo 12B) which mostly performs random guessing. This might indicate that coding models might better understand code correctness in their intrinsic preference.

**Comment distribution in positive/negative samples.** Our earlier finding in Table 5 shows that LLM-generated comments can be harmful to preference accuracy. To validate if this conclusion comes from a distribution match in the training and test sets, Table 11 shows the distribution of code comments in positive and negative samples from both training and test sets. We show that in the training set, positive samples overall have a bit more comments in Commit-Instruct-EditPack, suggesting they might have a bias to favor samples with more comments. However, we show that the positive samples in evaluation tend to have fewer code comments. This indicates that the negative impact of comments does not come from the distribution imbalance of code comments.

**Draft models and critic models.** While our *Critic-Evol* default setting uses a smaller draft model (8B) and a larger critic model (70B), Table 12 explores circumstances when using the same draft and critic models for synthesizing preference pairs. First, using the same draft and critic models leads to a higher filtering rate, meaning that more initial attempts are deemed "good enough" and thus not proceeding to the revision phase. This result is consistent with prior findings on LLM's self-bias (Xu et al., 2024; Li et al., 2024), *i.e.,* LLM judges tend to flavor their own generations. Meanwhile, there is a $2.5 \sim 9.4\%$ drop on the overall performance when using the same draft and critic models in *Critic-Evol*, yet it seems to be benign for the performance in the correctness category.

A.7   LIMITATION AND FUTURE WORK

While CODEFAVOR has demonstrated effectiveness in learning code preferences, there are several potential areas of improvement to enhance the scale, applicability, and accuracy of code preference models:

1. **Scaling up synthetic data:** One limitation in our implementation is the scale of synthetic training data, as our preliminary dataset only includes a total of 62,236 samples, which may be modest for model fine-tuning. Larger-scale datasets could further improve the generalizability and robustness of preference models for code generation. Since the idea of CODEFAVOR is rather general, we plan to scale up the synthetic data generation by collecting more code commits for *Commit-Instruct* and more LLM samples for *Critic-Evol*. Orthogonally, we may consider using multiple and more powerful models in *Commit-Instruct* and *Critic-Evol* to further improve the quality and diversity of generated synthetic data.

2. **Contextualized code preferences:** Code generation in real-world software development often involves broad context such as repository-level information (*e.g.,* (Ding et al., 2024; Zhang et al., 2023a)) and knowledge of external dependencies. Currently, CODEFAVOR focuses on code preferences of self-contained code snippets, which could limit the applications of code preference models in practically complex and context-dependent scenarios. Therefore, one future direction is to extend our framework to curate more context-sensitive code pairs for contextualized code preference learning.

3. **Benchmark improvements:** Our evaluation benchmark, CODEPREFBENCH, while carefully curated, also presents potential limitations related to the diversity and practicality of candidate code samples due to their synthetic nature. There may also be limitations due to the validity and consistency of human annotations, which are inherently subjective, particularly in assessing non-functional properties such as code efficiency. In the future, we aim to explore real-world preference data for evaluation and address challenges in human labeling through semi-automated strategies to supplement human assessments.

