# OpenReview forum: "Learning Code Preference via Synthetic Evolution"
_ICLR.cc/2025/Conference — Submitted to ICLR 2025_

### Official Review · Reviewer_cqfq · 2024-10-28

**Soundness:** 2
**Presentation:** 4
**Contribution:** 2
**Rating:** 6
**Confidence:** 5

**Summary:**

This paper aims to enable LLMs to better assess the quality of two code snippets by constructing a synthetic pairwise code preference dataset. The dataset is built using code commits (with pre- and post-commit versions as contrastive samples) and code critiques (the code snippet improved by a superior LLM as a contrastive sample). The authors have built benchmarks in Correctness, Efficiency, Security, and Human Preference to test the effectiveness of the proposed method.

**Strengths:**

- The synthetic dataset construction method (commits and critiques) is technically sound and novel to me.

- The authors conducted a comprehensive evaluation of the method. In addition to correctness, which is the focus of many traditional code generation studies, the authors also assess efficiency, security, and human developer preference.

- The authors put significant effort into the formatting of images and tables, which enhances the readability of the paper.

**Weaknesses:**

1. My main concern is that the authors overlook many works on code ranking and do not provide any experimental comparison. Many statements, such as "learning code preferences has been *largely under-explored*", "the *first* open recipe to train pairwise code preference models", and "understudied code generation domain", appear to overclaim. To name a few:

- A basic, training-free baseline is to compare the mean log probability of two code snippets and select the one with the highest probability, as in [1]. Furthermore, [2] also uses model likelihood for code selection.
- Some research also explores training classifiers to choose the better code, as in [3].
- The authors did not compare their work with the dataset in [4] mentioned in the Related Work section.
- In addition, but less importantly, to better contextualize the paper, some words about recent advances in execution-based code selection [5,6,7,8] would be appreciated. Particularly, [8] also employs a generation task similar to this paper. Considering that the work is recent, this comparison is not necessarily required.

Since the authors only reported the performance of the backbone LLMs and lacked empirical comparisons with advanced code selection methods, it is difficult to determine the relative improvement level of this work within related studies.

2. Some training details in the paper require further clarification. For instance, does the classifier task operate on the next token in Equation (1)? If so, considering that the label tokens ("A" or "B") and the criterion $c$ are tightly connected without a delimiter or explicit prompt, how does the LLM recognize where the criterion ends to output label tokens?

3. Since the authors collected both the training set and testing benchmarks, it's unclear whether they took decontamination steps to prevent test set leakage. If no decontamination was performed, analyzing the potential overlap between the training and test sets would be beneficial.

**Minor comments**

- The caption for Listing 1 is missing a period at the end.
- It would be better to place Equation (2) right after line 141.

**References**

[1] Evaluating Large Language Models Trained on Code, https://arxiv.org/abs/2107.03374

[2] Coder Reviewer Reranking for Code Generation, ICML 2023.

[3] Fault-Aware Neural Code Rankers, NeurIPS 2022.

[4] CodeUltraFeedback: An LLM-as-a-Judge Dataset for Aligning Large Language Models to Coding Preferences.

[5] CodeT: Code Generation with Generated Tests, ICLR 2023.

[6] Natural Language to Code Translation with Execution, EMNLP 2022.

[7] B4: Towards Optimal Assessment of Plausible Code Solutions with Plausible Tests, ASE 2024.

[8] Sifting through the Chaff: On Utilizing Execution Feedback for Ranking the Generated Code Candidates, ASE 2024.

**Questions:**

1. The Security score of human developers in Table 3 is only 59.7. Does this indicate that humans are not proficient at judging code security, even similar to random selection?

2. Could you further explain “Scores within 1 percentage point of the highest” in Table 3, as well as the detailed measurement method for "uncertain responses"?

3. The authors discovered that code comments may negatively affect model preferences, which is a bit strange and may be harmful to real-world applications. Is it possible to result from the class imbalance in comments (e.g., a higher proportion of comments in positive examples)? Could you provide the number of comments in positive and negative examples in the training and testing sets?

---

> ### Author Response · Authors · 2024-11-17
>
> > it's unclear whether they took decontamination steps to prevent test set leakage.
>
> Thanks for the question! We have carefully quantified the test set contamination in Appendix (A.5):
>
> * Figure 16 shows that **only 0.1% to 1.7%** of positive samples in the test-set code pairs can find training-set positive samples with a similarity score above 80.
> * As a reference, Riddell et al. (2024) show that 50.8% and 63.4% of code samples in the Stack (Li et al., 2023), can reach over 80 similarity scores with ground-truth code samples in MBPP (Austin et al., 2021) and HumanEval (Chen et al., 2021) respectively.
>
> > Could you provide the number of comments in positive and negative examples in the training and testing sets?
>
> Thanks for the question! Please kindly note that our empirical observation was that “LLM-generated” comments (not including human comments) may negatively impact LLM’s preference decision due to self-bias. We also run experiments to count the average number of comments between positive and negative samples:
>
> Training set: The following table lists the comment distribution in the two training sets. It shows that our observation is right – even if positive samples overall have a bit more comments, they do not seem to help CodeFavor models achieve better preference accuracy.
>
> | Training sets | Avg. Comments in Positive Code Samples | Avg. Comments in Negative Code Samples |
> | :---- | :---- | :---- |
> | Commit-Instruct-EditPack | 0.26 | 0.21 |
> | Critic-Evol-SOSS | 0.01 | 0.01 |
>
> Test-set: The following table shows the comment distribution of the **raw data** in our test-set – please note that our evaluation setup removes all comments when evaluating a model; therefore, imbalances in the raw data (if any) won’t impact the evaluation results presented in the paper.
>
> |  | Avg. Comments in Positive Samples | Avg. Comments in Negative Samples |
> | :---- | :---- | :---- |
> | Human Preference | 5.29 | 4.88 |
> | Code Correctness | 0.09 | 1.10 |
> | Code Efficiency | 0.0028 | 0.0028 |
> | Code Security | 0.93 | 0.94 |
>
> For the human preference, efficiency, and security categories, the average amount of comments is balanced. For the correctness category, comments amounts are unbalanced – after a closer investigation we found this is because (i) our code samples are generated from both base and instruction-tuned models; (ii) base models perform completion which preserves the docstring from the task description; (iii) instruction-tuned models tend not to repeat the docstring in the task description; (iv) instruction-tuned models are generally stronger than base models, leading to the imbalance.
>
> Our implementation to count the number of comments is listed below:
>
> ```python
> import re
>
> def count_comments(code_snippet):
>     single_line_comment_pattern = r"#.*"
>     multi_line_comment_pattern = r'"""(.*?)"""|\'\'\'(.*?)\'\'\''
>
>     # Find all single-line comments
>     single_line_comments = re.findall(single_line_comment_pattern, code_snippet)
>     # Find all multi-line comments
>     multi_line_comments = re.findall(multi_line_comment_pattern, code_snippet)
>     # Flatten multi-line comments and remove empty strings
>     multi_line_comments = [
>         comment for group in multi_line_comments for comment in group if comment
>     ]
>     return len(single_line_comments) + len(multi_line_comments)
> ```
>
> *(Author response to be continued in the next reply)*

---

> > ### Author Response · Authors · 2024-11-17
> >
> > > the authors overlook many works on code ranking and do not provide any experimental comparison…
> >
> > Thanks for bringing up all the related work! Our revision will definitely discuss them and compare ours with those that are applicable.
> >
> > Added new baseline results:
> >
> > |  | Correctness | Efficiency | Security | Avg |
> > | :---- | :---- | :---- | :---- | :---- |
> > | Logprob Sum (Mistral Nemo 12B)  | 28.6 | 52.0 | 61.4 | 47.3 |
> > | Logprob Sum (Llama 3.1 8B)  | 31.7 | 60.8 | 55.6 | 49.3 |
> > | (Best 8B Neural Scorer) Skywork-Reward-Llama-3.1-8B-v0.2 | 56.2 | 64.2 | 61.4 | 60.6 |
> > | (Ours as a reference)CodeFavor Llama-3-8B Classifier | 58.0 | 73.0 | 95.2 | 75.4 |
> >
> > * Decoding logprob/likelihood [1,2]: Following Codex, we compute the logprob sum and select the response with a higher logprob sum value as the preferred snippet. As we can see in the first two rows, the logprob-based approaches overall perform randomly.
> > * Regression-based neural rankers: We note that the “Fault-Aware Neural Code Rankers” work [3] did not release any model checkpoints. However, there have been various great reward models using similar modeling as Code Rankers (e.g., outputting a score). Therefore, we use “Skywork-Reward-Llama-3.1-8B-v0.2”, the **best** scoring-based reward model at its size from the Reward Bench Leaderboard (https://huggingface.co/spaces/allenai/reward-bench) as the baseline. The table above shows that our method is even better than the best general reward model by 24% at 8B among the code domain, even if the compared reward model is trained on a rich set of data generated by proprietary models such as GPT-4 and Claude-3-Opus.
> >
> > While we will definitely cite or have already cited the following work, they have been compared or are not applicable for comparison:
> >
> > * CodeUltraFeedback [4] prompts various LLMs to rate code snippets and construct preference pairs – in our evaluation, we also directly prompt LLMs as baselines to rank code pairs, which are already intrinsically compared with [4].
> > * As is earlier motivated in the paper, we focus on studying general scenarios, where test execution [5,6,7,8] is not always available (e.g., many code snippets cannot be executed/tested). Therefore, we only evaluate “static-analysis” based approaches including human baselines and models.
> >
> >
> >
> > > Could you further explain “Scores within 1 percentage point of the highest” in Table 3, as well as the detailed measurement method for "uncertain responses"?
> >
> > Thanks for pointing this out and we will make it more clear in our revision!
> >
> > * “Scores within 1 percentage point of the highest”: given all results in a list (say `scores`), we highlight the best results by making scores whose value is greater than `max(scores) - 1` bold. This highlighting helps readers quickly identify not just the single best result, but also other approaches that achieved nearly equivalent performance.
> > * "Uncertain responses": A human response is uncertain if all three annotations are “Tie” (e.g., the “Developer Agreement” block in Figure 12) or conflicting conclusions exist (Some choose “A”; while others choose “B”). A model response is uncertain when its response cannot be inferred as one response is better than the other regarding the mentioned criteria, for example, the “Gemini 1.5 Pro” block in Figure 13.
> >
> >
> > > The Security score of human developers in Table 3 is only 59.7. Does this indicate that humans are not proficient at judging code security, even similar to random selection?
> >
> > Thanks for the suggestion! The result answers our research question of “How good are generalist developers at differentiating secure and insecure code?” We conclude that they can be uncertain about code security most of the time (i.e., oftentimes “Tie”; yet this at least means they know when they don’t know).
> >
> > * Annotator background (Section 3.2): we aim to study general developers’ code preference accuracy and thus engage with 18 software engineers, most of which have more than two years of programming experience.
> > * Implication: Per the clarified background, the results indicate that generalist human developers might struggle with identifying code security issues in the code pairs we provided (See examples in A.4.3). Yet, 59.7 is still approximately 20% better than random guessing (50).
> >
> > > Some training details in the paper require further clarification.
> >
> > Thanks for the question! Indeed we should have provided more context.
> >
> > We followed the SliC-HF paper by Zhao et al (2023) and started our training based on instruction-tuned models. As such, given a user query $x$, the actual model received prompt will be templated according to the model's chat template and the special tokens in the chat template can split user queries and responses.
> >
> > We will include the additional explanation in our revision. Thank you!

---

> ### Comment · Reviewer_cqfq · 2024-11-17
>
> Thank you for your response; it has addressed many of my concerns.
>
> However, one of the baselines you compared is the Logprob **Sum**, which has proven highly inefficient in existing studies (See Figure 7 in https://arxiv.org/pdf/2107.03374) and performs even worse than random. A more reasonable comparison would be using the Logprob **Mean**, as the log prob sum tends to assign different scores to sequences with different lengths. This is also reflected in your reported results: Logprob Sum only achieves around 30% in correctness, significantly lower than random selection.
>
> In addition, what are the inputs for each baseline? Did they use the same prompt as your method, or only the code itself?

---

> > ### Author Response · Authors · 2024-11-17
> >
> > Thanks for the prompt response!
> >
> > > A more reasonable comparison would be using the Logprob Mean
> >
> > Thanks for the note! This is very helpful and we added the Logprob Mean results:
> >
> > |  | Correctness | Efficiency | Security | Avg |
> > | :---- | :---- | :---- | :---- | :---- |
> > | Logprob **Mean** (Llama 3.1 8B)  | 32.4 | 38.4 | 59.9 | 43.6 |
> > | Logprob Sum (Llama 3.1 8B)  | 31.7 | 60.8 | 55.6 | 49.3 |
> > | (Best 8B Neural Scorer) Skywork-Reward-Llama-3.1-8B-v0.2 | 56.2 | 64.2 | 61.4 | 60.6 |
> > | (Ours as a reference)CodeFavor Llama-3-8B Classifier | 58.0 | 73.0 | 95.2 | 75.4 |
> >
> > Overall we saw the Logprob mean result is similarly random and even leads to a worse overall result compared to Logprob Sum (probably just due to randomness). Note the results are computed by averaging logprob of all tokens (including prompt + response). If we mask prompts, and only consider response logprobs we get the following results, which are slightly better yet still somewhat random.
> >
> > |  | Correctness | Efficiency | Security | Avg |
> > | :---- | :---- | :---- | :---- | :---- |
> > | Logprob **Mean** (Llama 3.1 8B)  | 32.6 | 47.7 | 59.4 | 46.6 | |
> >
> > > what are the inputs for each baseline?
> >
> > Thanks for the question!
> >
> > * Logprob: we compute the score for each (prompt + response) applied with the corresponding model chat template. As such, we have a logprob-based score for each response (they share the same prompt) and we choose the response with the highest score.
> > * Skywork-Reward-Llama-3.1-8B-v0.2: Similarly to Logprob, we use (prompt + response) applied with the corresponding model chat template as the model's input -- getting two scores, selecting the best response.
> >
> > For a more detailed reference, below is the code to construct prompts for given a response pair:
> >
> > ```python
> >         prompts = [
> >             self.tokenizer.apply_chat_template(
> >                 [
> >                     {"role": "user", "content": prompt},
> >                     {"role": "assistant", "content": res},
> >                 ],
> >                 tokenize=False,
> >             ).replace(self.tokenizer.bos_token, "")
> >             for res in [resa, resb]
> >         ]
> > ```

---

> > > ### Comment · Reviewer_cqfq · 2024-11-21
> > >
> > > Thanks for the additional baselines and details. When an appropriate revised version of the manuscript is submitted, I tend to raise my score.

---

> > > > ### Author Response · Authors · 2024-11-22
> > > >
> > > > Thanks for the reply! We have updated the revised manuscript! Please let us know if you have further suggestions.

---

> > > > > ### Comment · Reviewer_cqfq · 2024-11-22
> > > > >
> > > > > Thank you for your revision and I have updated my score to 6. However, I noticed that your paper significantly exceeds the 10-page limit. I am unsure whether this complies with the guidelines.

---

> > > > > > ### Author Response · Authors · 2024-11-22
> > > > > > **Thank you!**
> > > > > >
> > > > > > Thank you for the generous reminder and score update!
> > > > > >
> > > > > > We have accordingly adjusted our revision to fit our main text in 10 pages.
> > > > > >
> > > > > > Cheers

---

### Official Review · Reviewer_zRJn · 2024-11-04

**Soundness:** 2
**Presentation:** 3
**Contribution:** 2
**Rating:** 5
**Confidence:** 4

**Summary:**

The paper proposes CODEFAVOR, a framework for training pairwise code preference models using synthetic evolution data generated from code commits and LLM critiques. This approach addresses the challenge of aligning code generation with developer preferences, focusing on correctness, efficiency, and security through a benchmark called CODEPREFBENCH, which includes 1,364 preference tasks. CODEFAVOR models achieve comparable performance to much larger models while being more cost-effective, and experiments reveal that human preferences often fall short in non-functional objectives like efficiency and security. The study provides insights into balancing model and human preferences, highlighting the potential limitations and strengths of each approach​.

**Strengths:**

* This paper contributes two code preference synthetic dataset and a CODEPREFBENCH, a collection of 1,364 carefully curated preference tasks, To evaluate code preferences labeled by various approaches.
* This paper comprehensively quantify and conduct case studies on code preferences derived from human developers and LLMs.
* CODEFAVOR models can match the preference accuracy of models that are larger by 6∼9×, while being cheaper by 34×

**Weaknesses:**

- The approach to synthetic data generation lacks originality, as creating datasets from git commits [1,6] and evolving from sampled code[2,3] are common practices in the field.
- The pairwise modeling approach is also not particularly novel; using pairwise prompts, criterion-based prompting, and classification or generation labels [4,5,7] have been previously explored in other studies.
- Additionally, there is concern that synthetic data generation may not fully ensure code correctness, as it heavily depends on the LLM used for critique and generation. The chosen model, Llama3-70B-Instruct, is relatively weak compared to state-of-the-art models and limited to only this single model.
- Finally, it is challenging to determine whether the performance gains following CODEFAVOR training are due to the distillation of knowledge from stronger LLMs used in data generation or from the CODEFAVOR training itself.


1. Jimenez, Carlos E., et al. "Swe-bench: Can language models resolve real-world github issues?." arXiv preprint arXiv:2310.06770 (2023).
2. Luo, Ziyang, et al. "Wizardcoder: Empowering code large language models with evol-instruct." arXiv preprint arXiv:2306.08568 (2023).
3. Wei, Yuxiang, et al. "Magicoder: Empowering code generation with oss-instruct." Forty-first International Conference on Machine Learning. 2024.
4. Dong, Yi, et al. "Steerlm: Attribute conditioned sft as an (user-steerable) alternative to rlhf." arXiv preprint arXiv:2310.05344 (2023).
5. Wang, Zhilin, et al. "Helpsteer: Multi-attribute helpfulness dataset for steerlm." arXiv preprint arXiv:2311.09528 (2023).
6. Ding, Yangruibo, et al. "Crosscodeeval: A diverse and multilingual benchmark for cross-file code completion." Advances in Neural Information Processing Systems 36 (2024).
7. Qin, Zhen, et al. "Large Language Models are Effective Text Rankers with Pairwise Ranking Prompting." Findings of the Association for Computational Linguistics: NAACL 2024. 2024.

**Questions:**

- Do you have concerns that the synthetic data generation methods, Commit-Instruct and Critic-Evol, may not fully ensure code correctness? If not, this raises another question: the quality of synthetic data is highly dependent on the LLM used to generate it. How do the experiments demonstrate that CODEFAVOR’s performance gains are due to the framework itself rather than simply distilling knowledge from a stronger LLM (Llama-3-70B-Instruct)?
- Could you provide more details on the inference process for the evaluation results in Table 3? Specifically, how many samples were created for each problem, what temperature was used, and are the results statistically significant?
- Could you elaborate on any aspects that emphasize the novelty of your work compared to previous studies?

---

> ### Author Response · Authors · 2024-11-17
>
> > How do the experiments demonstrate that CodeFavor’s performance gains are due to the framework itself rather than simply distilling knowledge from a stronger LLM (Llama-3-70B-Instruct)?
>
> Great question! Looking at Table 3 – the original Llama3-70B-Instruct’s overall score is 76.1 and our smaller models trained on data partially derived from Llama3-70B-Instruct can achieve a score of 77.7, which is even better than the derived model at a much smaller size. In theory, if we solely distill Llama3-70B-Instruct, it is unlikely to achieve a competitive score within a much smaller model parameter scale.
>
> Why is CodeFavor better than distillation? Please note that instead of solely distilling stronger models, our generated data is also derived from real-world code commits which brings additional information and weak labeling.
>
> > Could you provide more details on the inference process for the evaluation results in Table 3? Specifically, how many samples were created for each problem, what temperature was used, and are the results statistically significant?
>
> Thanks for raising the question! We use greedy decoding (mentioned in Section 3.1) for LLM generation and the prompt is listed in Listing 1 (mentioned in Appendix A.3). We implemented a post-processing method (Appendix A.3) to extract LLMs’ preferences from their generations. As we use greedy decoding, the result is theoretically deterministic.
>
> > Could you elaborate on any aspects that emphasize the novelty of your work compared to previous studies?
>
> Thanks for raising the concern. We argue that our paper is novel from various perspectives:
> * **Study-wise**, we invested significant efforts to study human annotation for code at scale and demonstrated various new insights.
> * **Benchmark-wise**, to our knowledge, CodePrefBench is the first benchmark to evaluate code preferences from models and non-model approaches, covering 4 detailed dimensions. We also provided a line of insights by conducting comprehensive evaluations.
> * **Technique-wise**, to our knowledge, we are the first work to construct synthetic code preference data, based on which we train competitive models to efficiently predict fine-grained code preference.
>
> > …creating datasets from git commits [1,6] and evolving from sampled code[2,3] are common practices in the field.
>
> While we greatly appreciate the reviewer for the question and references, we’d like to point out that our synthetic data generation has completely different purposes and implementations compared to the mentioned prior work.
>
> * We focus on **code preference training data**, which is a pair of code snippets where one is better than the other, alongside detailed criteria for comparison.
> * The mentioned work [2,3] synthesizes coding prompts and solutions, targeting the different applications of **code generation**. Meanwhile, the mentioned work [1,6] focuses on creating evaluation tasks rather than generating training data.
> * Meanwhile, we use the data (e.g., commits) differently: SWE-Bench [1] uses the pull request (which is also a commit) **partially** by **directly** using added test cases as an oracle. Our technique Commit-Instruct **fully** leverages the whole commit, including the pre-and post-commit code and the commit message, by rephrasing the noisy commits into a format that focuses on the actual change.
> * Lastly, the mentioned work [6] does not use nor mention git commit at all in their whole paper.
>
> > …synthetic data generation may not fully ensure code correctness….
>
> We agree that synthetic data can be low-quality if not well-created! We argue that our technique is empirically robust. This can be exemplified by Figures 3, 4, and 5 in the paper, where the preferred code snippets improve the quality of earlier code.
>
> Also please kindly note that our technique is not “pure distillation” and thus does not solely rely on LLM’s capability. For example, in Commit-Instruct, the synthesized code pair is inherited from code commits — empirically the post-commit code improves the pre-commit code; otherwise, they would have been filtered out.
>
> > Llama3-70B-Instruct is relatively weak compared to state-of-the-art models…
>
> Thanks for pointing this out!
>
> Please kindly note that our method can be applied to any model and in this paper, our choice of the critic model comes from legal reasons (Llama3 is license-friendly). Even though it's not the SOTA model, results still show significant improvements which proves that our method does not have to rely on proprietary models.

---

> > ### Author Response · Authors · 2024-11-22
> > **Gentle reminder 🤗**
> >
> > Dear reviewer,
> >
> > Thanks for your helpful questions and comments! We look forward to hearing your feedback and please do not hesitate to let us know if you have additional questions or concerns!
> >
> > Cheers

---

> > > ### Author Response · Authors · 2024-11-28
> > > **Reminder**
> > >
> > > Dear Reviewer,
> > >
> > > As the discussion period deadline approaches, we would appreciate your feedback on our response, new results, and revised manuscripts.
> > >
> > > Your feedback would greatly contribute to our work and the ICLR community!

---

> > > > ### Comment · Reviewer_zRJn · 2024-11-28
> > > >
> > > > I sincerely thank the authors for their efforts in clarifying the contributions and distinctions from prior works. While this is an intriguing new direction, the difference between code generation and code preference using commits seems relatively minor, as do the corresponding improvements in model performance. Furthermore, the proposed synthetic data generation method still appears to be limited to the scope of previously explored synthetic data generation methods.
> > > >
> > > > Although I will maintain the current score, I hope to see future iterations of this work that better demonstrate the advantages of code preference data over alternative approaches—either by showcasing stronger performance improvements or by presenting compelling use cases. Additionally, developing a novel synthetic data generation method that addresses key challenges, such as ensuring code functionality or preference correctness, could significantly enhance the contribution.

---

> > > > > ### Author Response · Authors · 2024-11-28
> > > > >
> > > > > We thank the reviewer for the reply.
> > > > >
> > > > > We respectfully disagree with the characterization of our work as involving "minor" adjustments to prior work, as our contributions required significant experimental design, innovation, and human resources. Below, we address specific points raised:
> > > > >
> > > > > ## > *"the difference between code generation and code preference using commits seems relatively **minor**"*
> > > > >
> > > > > We respectfully request clarification: can the reviewer point to any prior work on code generation that utilizes code commits for synthetic data generation in the manner we propose?
> > > > >
> > > > > We would like to reiterate and expand upon points from our earlier reply regarding related work:
> > > > >
> > > > > * Wizardcoder & Magicoder & Crosscodeeval: **NONE** of these approaches utilize code commits in any form.
> > > > > * Swe-bench: This work does **NOT** involve synthetic code generation; it uses GitHub issues (not commits) to construct evaluation tasks, which serve entirely different purposes and are unrelated to code instruction tuning.
> > > > >
> > > > > Given these distinctions, we argue that using code commits for synthetic data generation is a novel contribution. Furthermore, our application of this methodology to a new domain reinforces its originality.
> > > > >
> > > > >
> > > > > ## > *"as do the corresponding improvements in model performance."*
> > > > >
> > > > > ## > *"either by showcasing stronger performance improvements"*
> > > > >
> > > > > We would like to highlight that our approach improved model performance by up to **28%** across **ALL** evaluated models, ranging from 7B to 27B parameters. These results were achieved using fully permissive datasets and teacher models, without relying on proprietary models like GPT-4.
> > > > >
> > > > > Therefore, we don't think 28% is a number of "minor"; for example, the most cited paper, ResNet [1], improved prior work by 28% on the COCO dataset.
> > > > >
> > > > > [1] He et al. "Deep Residual Learning for Image Recognition"
> > > > >
> > > > >
> > > > > ## > *"developing a novel synthetic data generation method that addresses key challenges, such as ensuring code functionality or preference correctness, could significantly enhance the contribution."*
> > > > >
> > > > > While we thank the reviewers' thoughts on the new directions in this area, we believe it is important to assess our work based on what we have contributed, rather than what we have not done --- every paper has an infinite amount of "undone", but they still solve something.
> > > > >
> > > > > Please allow us to remind the reviewer of our notable contributions:
> > > > >
> > > > > 1. **Dimension & Benchmark:** We formulated the problem of "code preference learning" and built a benchmark of over a thousand evaluation tasks.
> > > > > 2. **Expensive human study:** We employed 18 developers to provide a human study on code preference. We showed a number of interesting findings on when human preference for code can be reliable.
> > > > > 3. **Technique:** For this new problem, we provide an effective model training recipe that can improve a given model's code preference accuracy by up to 28%. The technique suggests creating synthetic code preference pairs by weakly supervised data including code commits and strong LLM critiques over weak LLM generations.
> > > > >
> > > > > If this still cannot address the reviewer's concern,  we kindly request specific examples that can show what we have done is not enough or overlaps with prior publications. Thanks!

---

### Official Review · Reviewer_cc8R · 2024-11-06

**Soundness:** 2
**Presentation:** 3
**Contribution:** 3
**Rating:** 6
**Confidence:** 3

**Summary:**

The paper proposes CODEFAVOR, a framework for training pairwise code preference models from synthetic evolution data, including code commits and code critiques. To evaluate code preferences, the paper introduces CODEPREFBENCH, a benchmark comprising 1364 rigorously curated code preference tasks to cover three verifiable properties - correctness, efficiency, and security - along with human preference. The evaluation shows that CODEFAVOR holistically improves the accuracy of model-based code preferences.

**Strengths:**

(1) The paper is well written and easy to follow.

(2) The paper introduces a benchmark which can potentially be used by future papers.

(3) The developed approach is evaluated using multiple LLMs, showing that the developed approach is generally effective.

(4) The developed approach has good intuitions.

**Weaknesses:**

(1) In Table 1, it seems that the approaches in the rows are either LLMs, or LLMs with the training framework developed in this paper. To validate the effectiveness of the developed training framework, might it be helpful to add some baseline training approaches which also train the LLMs using the same training data used by CODEFAVOR?

(2) In Table 1, considering that there is still a gap between the Open-Weight Models and Our Models and Baselines (i.e., LLMs used with CODEFAVOR), might it be helpful to understand whether CODEFAVOR can further improve these Open-Weight Models with larger sizes?

(3) It might be helpful if the paper can show the effectiveness of the developed approach when the approach is applied to some other LLMs for code related tasks (e.g., Code Llama, Lemur).

**Questions:**

(1) In Table 1, to validate the effectiveness of the developed training framework, might it be helpful to add some baseline training approaches which also train the LLMs using the same training data used by CODEFAVOR?

(2) In Table 1, might it be helpful to understand whether CODEFAVOR can further improve these Open-Weight Models with larger sizes?

(3) Would the developed approach also be effective, if the developed approach is applied to some other LLMs for code related tasks (e.g., Code Llama, Lemur)?

---

> ### Author Response · Authors · 2024-11-17
>
> > To validate the effectiveness of the developed training framework, might it be helpful to add some baseline training approaches which also train the LLMs using the same training data used by CODEFAVOR?
>
> Thanks for the question! Please kindly note that our framework is a combination of (i) training data generation; and (ii) model training scheme – the effectiveness is demonstrated through the co-design of both components.
>
> Our understanding of the reviewer’s question is to study the (ii) component, i.e., the model training schemes (Please kindly let us know if we misunderstood).
>
> * In Table 5 from the paper, we thoroughly studied various training schemes including classification, generation, and model merging.
> * To further improve the study thoroughness, we follow the RLHF literature and additionally trained a Bradley–Terry model (Dong et al, 2024) for comparison: Surprisingly, while the overall performance of Bradley-Terry modeling is suboptimal to the classification modeling (4% weaker), its preference accuracy on code correctness beats all evaluated LLMs approaches including Llama-3.1-405B-Instruct.
>
> |  | Correctness | Efficiency | Security | Avg |
> | :---- | :---- | :---- | :---- | :---- |
> | CodeFavor Llama-3-8B Classification | 58.0 | 73.0 | 95.2 | 75.4 |
> | CodeFavor Llama-3-8b Bradley-Terry  | 75.0 | 59.7 | 82.6 | 72.4 |
>
> Dong et al. RLHF Workflow: From Reward Modeling to Online RLHF. TMLR, 2024.
>
> > In Table 1… whether CODEFAVOR can further improve these Open-Weight Models with larger sizes?
>
> Great suggestion! With our best computing budget (8xA100@80G), we additionally trained a CodeFavor classifier based on Gemma2-27B-IT with data mixture.
>
> |  | Correctness | Efficiency | Security | Avg. |
> | :---- | :---- | :---- | :---- | :---- |
> | Gemma2-27B (Baseline) | 55.4 | 78.4 | 80.8 | 71.5 |
> | CodeFavor Gemma2-27B (Classification) | 65.6 | 73.0 | 96.1 | 78.2 |
> | CodeFavor Gemma2-9B (Classification) | 56.8 | 75.3 | 92.3 | 74.8 |
>
> We show that:
>
> * Our method can further improve the larger 27B open-weight models by 9.4%.
> * Meanwhile, compared to the 9B CodeFavor model, the 27B CodeFavor classifier can further improve it by ~5%.
>
> >  … effectiveness … when the approach is applied to some other LLMs for code related tasks (e.g., Code Llama, Lemur).
>
> Thanks for the great suggestion! We extended the new experiments below by training CodeLlama 13B based on CodeFavor and were able to achieve an overall improvement of 16-17%.
>
> We are especially grateful for this suggestion as we found CodeLlama achieved a positive default correctness score without tuning compared to other general models (such as Mistral Nemo 12B) which mostly randomly guess. This might indicate that code instruct models might have a better sense of code correctness in their intrinsic preference.
>
> |  | Correctness | Efficiency | Security | Avg. |
> | :---- | :---- | :---- | :---- | :---- |
> | CodeLlama 13B Instruct (Baseline) | 57.3 | 64.3 | 74.9 | 65.5 |
> | CodeLlama 13B (CodeFavor Classifier) | 57.7 | 73.3 | 96.6 | 75.9 |
> | CodeLlama 13B (CodeFavor Generator) | 59.5 | 78.1 | 92.3 | 76.6 |

---

> > ### Author Response · Authors · 2024-11-22
> > **Gentle reminder 🤗**
> >
> > Dear reviewer,
> >
> > Thanks for your helpful questions and comments! We look forward to hearing your feedback and please do not hesitate to let us know if you have additional questions or concerns!
> >
> > Cheers

---

> > > ### Author Response · Authors · 2024-11-28
> > > **Reminder**
> > >
> > > Dear Reviewer,
> > >
> > > As the discussion period deadline approaches, we would appreciate your feedback on our response, new results, and revised manuscripts.
> > >
> > > Your feedback would greatly contribute to our work and the ICLR community!

---

### Official Review · Reviewer_4Aq4 · 2024-11-09

**Soundness:** 2
**Presentation:** 4
**Contribution:** 3
**Rating:** 5
**Confidence:** 4

**Summary:**

The paper addresses the challenge of assessing code generation based on well-formed properties and aligning it with developer preferences, which has proven difficult in the context of Large Language Models (LLMs). To tackle this issue, the authors propose CODEFAVOR, a framework designed to train pairwise code preference models using synthetic evolution data, including code commits and critiques. Additionally, they introduce CODEPREFBENCH, a benchmark consisting of 1364 curated code preference tasks that evaluate three key properties: correctness, efficiency, and security, alongside human preferences. The main results indicate that CODEFAVOR significantly enhances the accuracy of model-based code preferences by up to 28.8%, while also demonstrating that these models can perform comparably to those with 6 to 9 times more parameters, all while being 34 times more cost-effective. Furthermore, the study highlights the limitations of human-based code preference assessments, revealing that a substantial percentage of tasks remain unsolved despite considerable time investment.

**Strengths:**

This paper presents a significant advancement in the field of code preference learning by introducing the CODEFAVOR framework, which innovatively utilizes synthetic evolution data to train models that predict meaningful code preferences. The novelty lies in its dual focus on aligning human and model preferences with verifiable code properties, addressing a critical gap in existing research. Key contributions include the development of CODEPREFBENCH, a comprehensive benchmark with 1364 curated tasks that evaluate code based on correctness, efficiency, and security, thus providing a robust evaluation framework for future studies. The results demonstrate that CODEFAVOR can enhance model accuracy by up to 28.8% while being more cost-effective than larger models, highlighting its practical significance in improving code generation assessments. Additionally, the paper sheds light on the limitations of human-based assessments, emphasizing the need for model-based approaches in evaluating non-functional code properties, which further underscores the importance of the research findings.

**Weaknesses:**

The problem formulation/setting can be improved in terms of clarity, motivation, and realism. The framework is proposed to serve code assessment purposes, i.e., judging automatically which version of code generated by a model from a prompt is preferred (i.e. more correct/secure/efficient) between a pair of two versions. The questions are (1) in what scenarios would these two versions be available, and (2) how realistic it is that there are such strong and discriminative contrasts between the two versions (i.e., correct versus wrong, fast versus slow, secure versus vulnerable). In a typical use scenario of LLMs for code generation, developers may feed the LLM with a prompt and get a response. Should they always ask the model for two responses? If so, the cost would double. More importantly, it is probably unlikely that there is a such contrast between the two versions of code generated from the same prompt---e.g., the two versions could be similarly good or bad. Learning preferences in a strongly differentiable pair of two responses does not seem to be realistic. If so, the paper may want to provide supporting evidence that this is the case. Or the problem itself is not motivated convincingly.

Another primary weakness is the heavy reliance on synthetic evolution data for training the CODEFAVOR framework. While synthetic data can be useful, it may not fully capture the complexities and nuances of real-world coding scenarios. This limitation raises concerns about the generalizability of the model's performance in practical applications, as the evaluation may not reflect actual developer preferences or code behavior in diverse environments.

The paper acknowledges the prohibitive costs and limitations of human-based code preference assessments, noting that despite significant time investment, a substantial percentage of tasks remain unsolved (15.1% to 40.3%) . This suggests that human evaluators may struggle with certain tasks, which could undermine the reliability of the human preference data used for comparison. The paper could benefit from a more in-depth exploration of these limitations and their implications for the overall findings.

The paper mentions that the reliability of using large language models (LLMs) as evaluators often hinges on their reasoning capabilities, which can be subject to inherent biases . This raises questions about the objectivity of the model-based preferences derived from LLMs, as biases could skew the results and affect the alignment with human preferences. A more thorough examination of potential biases and their impact on the findings would strengthen the paper's arguments.

The evaluation framework, CODEPREFBENCH, focuses on three specific properties: correctness, efficiency, and security. While these are important aspects, the paper does not justify the choices, among various other quality aspects of code, such as maintainability or readability. Also, it seems that each of these chosen properties is separately considered, yet in real-world scenarios developers need to balance multiple factors at the same time when choosing which code to adopt (e.g., code that is both secure and correct). The interplay among these potentially competing factors is not considered in the approach nor in the evaluation.

**Questions:**

Q1: what are the backgrounds of the developers participating in the evaluation and data annotation, and how their potential biases may have affected the soundness of the approach and the evaluation?

Q2: the larger LLMs have clear edges over the CodFavor improved small models with much lower costs than human evaluators. Why would they not be a better option than using CodeFavor to improve the smaller models?

---

> ### Author Response · Authors · 2024-11-17
>
> > what are the backgrounds of the developers participating in the evaluation and data annotation, and how their potential biases may have affected the soundness of the approach and the evaluation?
>
> Thanks for the question! Section 3.2 provides a detailed description of the annotators’ backgrounds: “Our annotation team consists of 18 software developers, two-thirds of which hold degrees in computer science, and 95% of them have over two years of programming experience. For Python proficiency, 43% of them self-rate as advanced, while the rest consider themselves middle-level.”
>
> Specifically, we mitigate potential individual biases with our best efforts:
>
> 1. We engage with as many as 18 developers
> 2. We have each task annotated by three different developers through major voting
> 3. We comprehensively evaluate over a thousand tasks
>
> Notably, to ensure the quality of human evaluation, we iteratively refined the annotation guidelines, carefully communicated with the annotators, and performed post-annotation inspection.
>
> > …the larger LLMs have clear edges over the CodeFavor-improved small models... Why would they not be a better option than using CodeFavor to improve the smaller models?
>
> Users should always use CodeFavor models in place of the raw model as CodeFavor can improve the preference accuracy of both small and large models.
>
> *Our existing experiments have shown that it achieves up to 28% improvement for 8-12B models.
> * Additionally, with our best computing budget, we apply CodeFavor to Gemma2-27B-IT (data mixture), where an improvement is also observed by as much as 9%:
>
>
> |  | Correctness | Efficiency | Security | Avg. |
> | :---- | :---- | :---- | :---- | :---- |
> | Gemma2-27B-IT (Baseline) | 55.4 | 78.4 | 80.8 | 71.5 |
> | Gemma2-27B-IT \+ CodeFavor (Classification) | 65.6 | 73.0 | 96.1 | 78.2 |
>
> We initially focused on small models as they are more efficient to train and deploy.
>
> > The paper mentions that the reliability of using large language models (LLMs) as evaluators often hinges on their reasoning capabilities… A more thorough examination of potential biases and their impact on the findings would strengthen the paper's arguments.
>
> Thoroughly studying the LLM evaluators’ bias is definitely important!
>
> That is exactly why in Appendix (A.4) we put nearly 10 pages of case studies and summaries to understand the preference and bias patterns of evaluated models. For example, Gemini 1.5 Pro seems reluctant to sharpen its preference for security-related code snippets. We hope our analysis can provide insights and references for future research.
>
> > … the prohibitive costs and limitations of human-based code preference assessments … suggests that human evaluators may struggle with certain tasks... The paper could benefit from a more in-depth exploration of these limitations and their implications for the overall findings.
>
> Thanks for the suggestion! At a high level, our paper explores human-based code preference in various dimensions (Section 3.2), including expertise, confidence, overhead, etc. Additionally, at a lower level, in Appendix A.4, we provide 10 case studies that compare model and human behaviors in detail.
>
> Overall we found:
> 1. Human preference for code is expensive and slow
> 2. Human preference strongly aligns with **code correctness** but can struggle with **non-functional properties**
> 3. The 2nd conclusion can come from the fact that generalist developers are familiar with writing test cases to validate code but they might not have enough background in code optimization and code security, suggesting that domain experts or LLM assistance are generally needed when assessing non-functional code properties
>
> > in what scenarios would these two versions be available?
>
> Great question! Accurate code preference can be applied in various scenarios including (i) quality assurance of code commits, where we detect if post-commit code is better than pre-commit one; (ii) inference-time sample ranking; (iii) training reward models (i.e., our experimental setting); and (iv) provide data and signal for preference optimization algorithms such as DPO.
>
> > While synthetic data can be useful, it may not fully capture the complexities and nuances of real-world coding scenarios.
>
> Thanks for the comment! Maintaining the real-world complexities in synthetic data is definitely important! That is exactly why we specifically leveraged weak supervision in data generation rather than performing pure distillation:
>
> * In Commit-Instruct, the synthesized data comes from rephrasing diverse, **real-world** commits collected and cleaned carefully from GitHub to make sure the training data reflects real-world code usage.
> * Beyond human-created commits, we also want to cover code that is directly generated by LLMs, given that nowadays lots of GitHub code can be partially generated by LLMs. As such, Critic-Evol constructs code preference pairs by capturing and fixing code defects generated by smaller models.

---

> > ### Author Response · Authors · 2024-11-22
> > **Gentle reminder 🤗**
> >
> > Dear reviewer,
> >
> > Thanks for your helpful questions and comments! We look forward to hearing your feedback and please do not hesitate to let us know if you have additional questions or concerns!
> >
> > Cheers

---

> > > ### Author Response · Authors · 2024-11-28
> > > **Reminder**
> > >
> > > Dear Reviewer,
> > >
> > > As the discussion period deadline approaches, we would appreciate your feedback on our response, new results, and revised manuscripts.
> > >
> > > Your feedback would greatly contribute to our work and the ICLR community!

---

### Author Response · Authors · 2024-11-21
**Meta Response & Submission Revision**

We thank the reviewers for the inspiring comments and we have updated our submission by making the following notable changes, highlighted using purple in the updated submission PDF:

* **(Experiment) CodeFavor for larger models (Appendix A.6 / Table 9):** We applied CodeFavor to a much larger model, namely Gemma2-27B-IT, showing an overall improvement of 9%. (4Aq4, cc8R)
* **(Experiment) Adding Bradley-Terry modeling as a baseline (Appendix A.6 / Table 10):** Using BT modeling, it achieves a suboptimal overall result compared to classification/generation, but it can achieve a substantially better correctness score specifically (cc8R)
* **(Experiment) Applying CodeFavor to CodeLlama (Appendix A.6 / Table 11):** CodeFavor improves coding models also quite well; the untuned coding model achieves a better correctness score compared to general models of similar sizes (cc8R)
* **(Experiment) Comment distribution in positive & negative samples (Appendix A.6 / Table 12):** The results show that the negative impact of comments does not simply come from distribution imbalance in training/evaluation sets (cqfq)
* **(Experiment) Comparing against Logprob Mean & code scoring models (Table 3):** logprob mean leads to rather random results and CodeFavor models largely outperform the leading 8B general reward model (cqfq)
* **(Discussion) Acknowledge missed prior work** (zRJn, cqfq)
* **(Writing) Clarifying training details (Section 2.1):** prompt and output token are separated by special tokens defined by the chat template (cqfq)

We look forward to hearing further feedbacks from the reviewers!

---

### Author Response · Authors · 2024-11-25
**Gentle reminder of reviewer responses**

Dear Reviewers,

As the discussion period deadline approaches, we would appreciate your feedback on our response. Your feedback would greatly contribute to our work and the ICLR community.

Thank you for your time and consideration!

---

### Author Response · Authors · 2024-12-02

Dear Reviewers 4Aq4, cc8R, and zRJn:

We'd like to gently remind you that December 2nd (end of day in AoE) is the last day for reviewer feedback and we can reply to additional questions until tomorrow. Please kindly take a look at our replies and updated manuscript with extensive additional experiments and reference updates, and let us know if your concerns have been addressed.

Thank you for your time!

---

### Meta-Review · Area_Chair_WTn3 · 2024-12-23

**Metareview:**

This paper is on the problem space of code generation using LLMs with a focus on aligning the code with developer preferences. The paper develops an approach for training pairwise code preference models using synthetic data and introduces a benchmark that considers correctness, efficiency, and security along with human preferences. Experimental evaluation shows improvement in accuracy of code preference model.

The reviewers' appreciated the importance of this research problem and contributions on both benchmark and preference model. They also asked a number of questions ranging from motivation to problem formulation and experimental methodology. Some of the questions were answered by the author rebuttal. There are two important outstanding concerns that were raised by Reviewer 4Aq4 which were not addressed.
1. Motivation is not strong and/or unclear.
2. The realistic use-case of this study from a software developers' point of view.

Therefore, I'm recommending to reject this paper and strongly encourage the authors' to improve the paper based on the feedback from reviewers' for resubmission.

**Additional Comments On Reviewer Discussion:**

Summarized in the meta review.

---

### Decision · Program_Chairs · 2025-01-22

Reject